# VideoHallu: Evaluating and Mitigating Multi-modal Hallucinations on Synthetic Video Understanding

**Zongxia Li**[†][*]    **Xiyang Wu**[†][*]    **Guangyao Shi**[‡]    **Yubin Qin**[†]    **Hongyang Du**[†]

**Tianyi Zhou**[†]    **Dinesh Manocha**[†]    **Jordan Lee Boyd-Graber**[†]

[†]University of Maryland, College Park    [‡]University of Southern California

`{zli12321, wuxiyang, Yubinq, hydu, zhou, dmanocha, ying}@umd.edu,`
`shig@usc.edu`

## Abstract

Vision–Language Models (VLMs) have achieved remarkable success in video understanding tasks. Yet, a key question remains: do they comprehend visual information, or merely learn superficial mappings between visual and textual patterns? Understanding visual cues, particularly those related to physics and common sense, is crucial for AI systems interacting with the physical world. However, existing VLM evaluations primarily rely on *positive-control* tests using real-world videos that resemble training distributions. While VLMs perform well on such benchmarks, it is unclear whether they grasp underlying visual and contextual signals or simply exploit visual-language correlations. To fill this gap, we propose incorporating *negative-control* tests, *i.e.*, videos depicting physically impossible or logically inconsistent scenarios, and evaluating whether models can recognize these violations. True visual understanding should evince comparable performance across both positive and negative tests. Since such content is rare in the real world, we introduce VideoHallu, a synthetic video dataset featuring physics- and commonsense-violating scenes generated using state-of-the-art tools such as Veo2, Sora, and Kling. The dataset includes expert-annotated question–answer pairs spanning four categories of physical and commonsense violations, designed to be straightforward for human reasoning. We evaluate several leading VLMs, including Qwen-2.5-VL, Video-R1, and VideoChat-R1. Despite their strong performance on real-world benchmarks (*e.g.*, MVBench, MMVU), these models hallucinate or fail to detect physical or logical violations, revealing fundamental weaknesses in visual understanding. Finally, we explore reinforcement learning–based post-training on our *negative* dataset: fine-tuning improves performance on VideoHallu without degrading results on standard benchmarks—indicating enhanced visual reasoning in VLMs. Our data is available at https://github.com/zli12321/VideoHallu.git.

## 1 Introduction

Vision–Language Models (VLMs) have made remarkable progress in video understanding. However, they remain prone to hallucinations and shallow visual reasoning [1–3]. Prior works mitigate these issues across various domains, including chart interpretation [4], video understanding [5], and visual question answering (VQA) [6], primarily through supervised fine-tuning (SFT) or R1-style chain-of-thought training (reinforcement learning) [7, 8]. However, most of these VLM evaluations rely on *positive-control* test, that is, real-world data drawn from distributions closely aligned with training data. Consequently, it remains unclear whether current VLMs genuinely reason about visual cues or merely exploit prior visual-language correlations within familiar distributions [9].

To truly evaluate visual understanding, mwe test VLMs under *negative-control* conditions, *i.e.*, videos outside their training distribution that depict physically impossible or logically inconsistent events.

---

[*]Equal contribution.

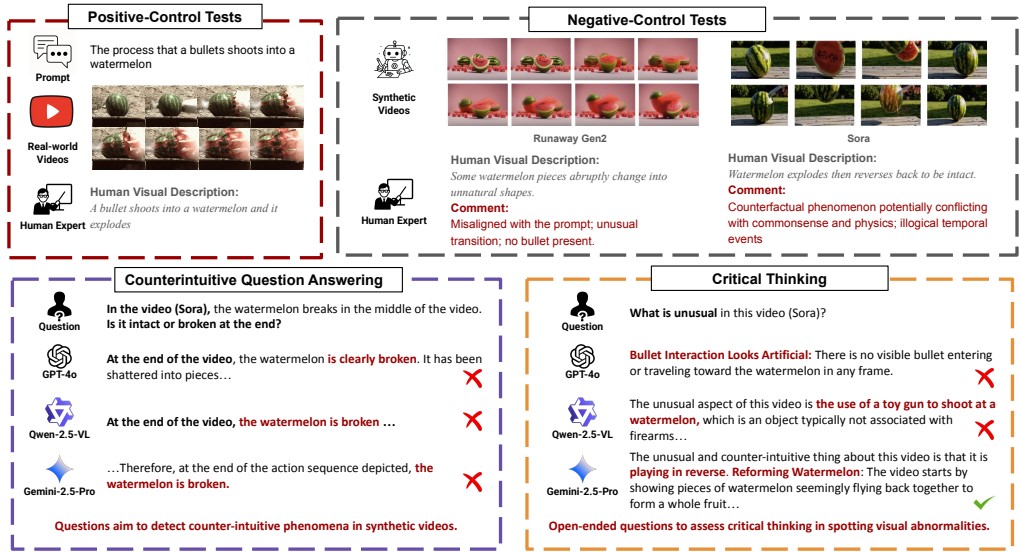

Figure 1: **Illustrative examples of designed negative-control tests to evaluate the critical thinking abilities of VLMs.** Unlike real-world videos, synthetic videos can contain counterfactual or commonsense-violating contexts misaligned with reality. VideoHallu includes such synthetic videos with perceptually obvious abnormalities, paired with crafted questions that probe counterintuitive phenomena or test VLMs' critical thinking in detecting such abnormalities. When SOTA VLMs are evaluated on VideoHallu, they frequently hallucinate, which suggests that these models rely on language priors and commonsense knowledge rather than truly understand the videos.

These tests reveal whether models detect violations of physics, causality, or commonsense instead of relying on memorized language knowledge. However, constructing such out-of-distribution (OOD) videos in the real world is costly and impractical [10].

Modern video generation models such as Veo2, Sora, and Runway [11–14] can produce photorealistic but physically impossible scenes. Such models provide an alternative option to generate test videos for probing VLMs' visual understanding. By careful design, these synthetic videos can be systematically introduced to include violations of gravity, causality, and commonsense interactions [15, 16], enabling controlled OOD evaluations where models need to rely on visual cues. Current VLMs, predominantly trained on videos conforming to physical laws, may thus overfit to statistical regularities rather than learning genuine causal reasoning [17]. Figure 1 illustrates that even state-of-the-art VLMs such as Gemini-2.5-Pro [18], GPT-4o [19], and Qwen2.5-VL [20] hallucinate when confronted with counterintuitive scenarios. For instance, when a watermelon reassembles after an explosion, models rely on linguistic priors (*e.g.*, *"a watermelon should break when shot"*) rather than actual visual cues, exposing their limited physical reasoning.

To rigorously evaluate such limitations, we introduce VideoHallu, a dataset of expert-curated question–answer pairs over synthetic videos featuring controlled violations of alignment, spatial–temporal consistency, commonsense, and physics. We benchmark several leading VLMs and analyze their failure modes on these physically and logically inconsistent videos. Finally, we explore two post-training strategies, supervised fine-tuning (SFT) and reinforcement learning (RL) via Group Relative Policy Optimization (GRPO) [21], using both real-world data from Video-R1 [7] and synthetic data from VideoHallu. GRPO enhances generalization on synthetic video reasoning without degrading real-world performance.

**Contributions.**

**1**) We introduce VideoHallu, a dataset of 3K expert-annotated QA pairs on synthetic videos that include violations spanning alignment, consistency, commonsense, and physical reasoning.

**2**) We evaluate state-of-the-art VLMs and find that even top-performing models (*e.g.*, GPT-4o, Gemini-2.5-Pro) can achieve only ∼50% accuracy in our dataset, exhibiting frequent hallucination in counterintuitive scenarios.

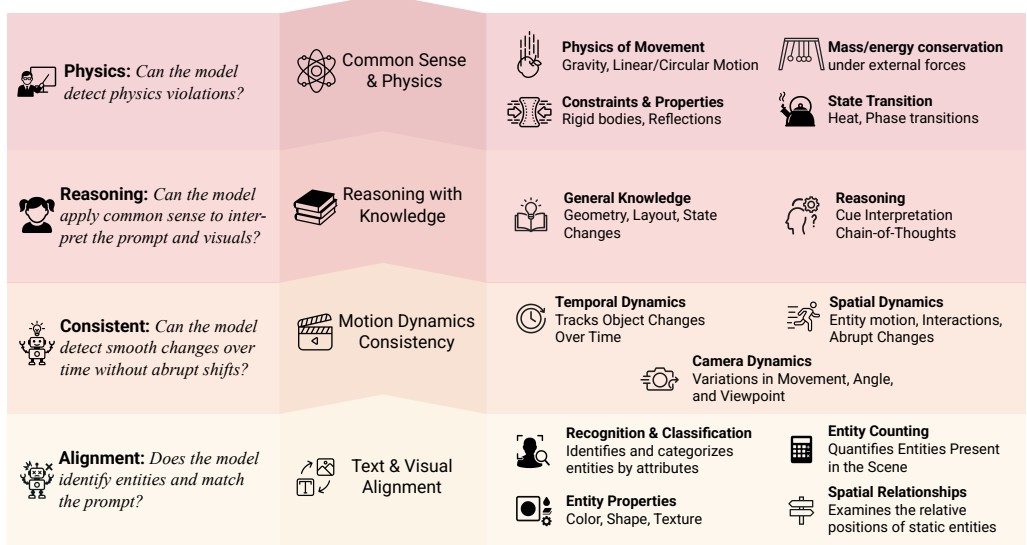

Figure 2: **Question Categorization of VideoHallu.** We design our benchmark, VideoHallu, with four question categories to probe limitations in synthetic video understanding, covering perceptual understanding to abstract reasoning: **(a) Physics** assesses if the model applies physical laws to entity motions and procedural understanding. **(b) Common Sense Reasoning** tests if the model can reason based on its knowledge. **(c) Spatial-temporal Consistency** examines whether the model can track entity motion across frames. **(d) Alignment** checks if the model correctly identifies and understands entities using visual and textual cues.

**3**) GRPO post-training with synthetic data improves visual reasoning on VideoHallu while maintaining real-world performance, providing a path toward more physically grounded VLMs.

## 2 VideoHallu: Evaluating VLMs' Synthetic Video Understanding

**Preliminary.** Our objective is to evaluate whether VLMs can effectively reason about and answer questions concerning synthetic videos that fall outside the distribution of their training data. To construct such evaluation data, we synthesize videos from text prompts using generative models. For these synthetic test videos to be meaningful, they should incorporate deliberate visual abnormalities that elicit responses contradicting common-sense expectations—situations in which a model relying solely on linguistic reasoning would produce one answer, but where careful visual observation reveals an alternative, visually grounded truth. To see how VLMs handle synthetic videos with such *abnormalities*, we categorize our evaluation questions into *counter-intuitive* and *critical thinking* types. Counter-intuitive questions focus on implausible or physically impossible events (*e.g.*, a shattered watermelon reassembling itself), while critical thinking questions evaluate the model's ability to detect visual inconsistencies or logical contradictions (*e.g.*, unnatural object breakage).

**Data Collection.** Our data collection pipeline consists of two main stages: The first stage generates synthetic videos $V$ with common sense or physics abnormalities, *i.e.*, videos that satisfy constraints (5), where the LLM backbone possesses human-aligned knowledge but VLMs overlook abnormalities, resulting in answers that disagree with human perception. We recruited five human experts to review the defined abnormality categories (detailed in Table 2 and Appendix C) and craft prompts that reproduce such abnormalities in generated synthetic videos. In total, they created 141 adversarial prompts, used to generate 987 videos across seven models: Sora [22], Veo2 [11], Runway Gen 2 [13], Kling [23], Pixverse [24], Lavie [25], and CogVideo [26].

In the second stage, we craft adversarial video QA pairs to evaluate VLMs' understanding of synthetic videos. Human experts manually review each video to identify counterintuitive contexts that lead to significant discrepancies between VLM outputs and human perception, *i.e.*, video QA pairs maximizing the objective function (4). They then construct natural language questions—along with

the ground truth answer—based on the context. These QA pairs are categorized into sub-categories (Table 2). Each annotator writes QA pairs highlighting visually clear but semantically abnormal content, difficult for VLMs to detect. These questions are not designed to trick models but rather to probe their ability to catch subtle violations of common sense, physics, or prompt-video mismatches, critical for robust, interpretable video evaluation (Figure 3).

**Dataset Metadata.** Our dataset comprises 3,233 video question–answer pairs with no video overlap across splits: 800 pairs for training, 908 for validation, and 1,525 for testing. The videos average 96.0 frames per video corresponding to approximately 5.3 seconds at an average framerate of 23 FPS. Frame resolution averages 1042 × 588 pixels across all videos.

# 3 Experiment and Results

Given the collected adversarial QA pairs, we evaluate 17 SOTA VLMs (Table 1). For models not trained with RL or chain-of-thought (CoT) generation, we use standard prompting to generate direct answers. For those trained with RL or CoT supervised finetuning (*e.g.*, Video-R1-CoT [7] and VideoChat-R1-think [8]), we prompt them to generate step-by-step critical thinking and reasoning before generating a final answer (Appendix. D). Figure 3 highlights hallucinations produced by SoTA models across all four categories in synthetic video understanding tasks, with the hallucinated contexts marked within each answer (additional examples in Appendix A).

**Answer Evaluation:** We adopt LLM-as-a-Judge [27–29] as our evaluation method. GPT-4o-mini evaluates the correctness of model responses (§ 6).[2]

| Model | Alignment | Physics | Consistency | Commonsense | Overall |
|-------|-----------|---------|-------------|-------------|---------|
| *VLMs: <7B* | | | | | |
| SmolVLM-3B [30] | 15.94 | 13.44 | 22.49 | 8.75 | 16.13 |
| Qwen2.5-VL-3B [20] | 41.53 | 27.21 | 26.91 | 26.25 | **35.48** |
| InternVL3-2B [31] | 47.36 | 32.79 | 42.17 | 32.50 | 42.82 |
| *VLMs: >7B* | | | | | |
| LLaVA-OneVision [32] | 44.22 | 32.46 | 32.13 | 45.00 | 39.93 |
| Video-LLaVA [33] | 46.58 | 40.00 | 43.37 | 31.25 | 43.93 |
| LLaVA-NeXT [34] | 50.95 | 36.07 | 38.96 | 31.25 | 44.98 |
| Video-LLaMA [35] | 55.67 | 38.69 | 50.20 | 32.50 | 50.16 |
| InternVL3-9B [31] | 53.54 | 43.61 | 47.79 | 38.75 | 49.84 |
| InternVL3-14B [31] | 53.65 | 45.90 | 46.18 | 31.25 | 49.70 |
| InternVL3-38B [31] | 55.78 | 38.69 | 50.20 | 38.75 | 50.56 |
| Qwen2.5-VL-32B [20] | 58.81 | 42.95 | 46.59 | 40.00 | 52.66 |
| Qwen2.5-VL-7B [20] | 58.02 | 44.59 | 46.99 | 47.50 | **52.98** |
| *VLMs: R1-finetuned* | | | | | |
| VideoChat-R1 [8] | 53.31 | 40.33 | 44.58 | 45.00 | 48.85 |
| Video-R1-SFT [7] | 58.14 | 47.21 | 48.19 | 41.25 | 53.44 |
| Video-R1 [7] | 58.14 | **48.20** | **49.00** | 38.75 | **53.64** |
| *VLMs: Close-Source* | | | | | |
| Gemini-2.5-Pro [18] | 58.36 | 33.11 | 40.16 | 36.25 | 49.18 |
| Gemini-2.0-Flash [36] | 56.57 | 39.02 | 42.97 | 40.00 | 49.97 |
| GPT-4o-mini [19] | 54.88 | 41.97 | 48.19 | 38.75 | **50.36** |

Table 1: **Video Model Evaluation Results.** We evaluate diverse VLMs across different sizes on our test set, reporting alignment, physics understanding, spatial-temporal consistency, and commonsense reasoning. Video-R1 is best overall.

---

[2]For CoT generations, we extract the final answer as responses to evaluate. To validate reliability on our dataset, we manually annotated 200 randomly sampled answer pairs, achieving 97% agreement with GPT-4o-mini.

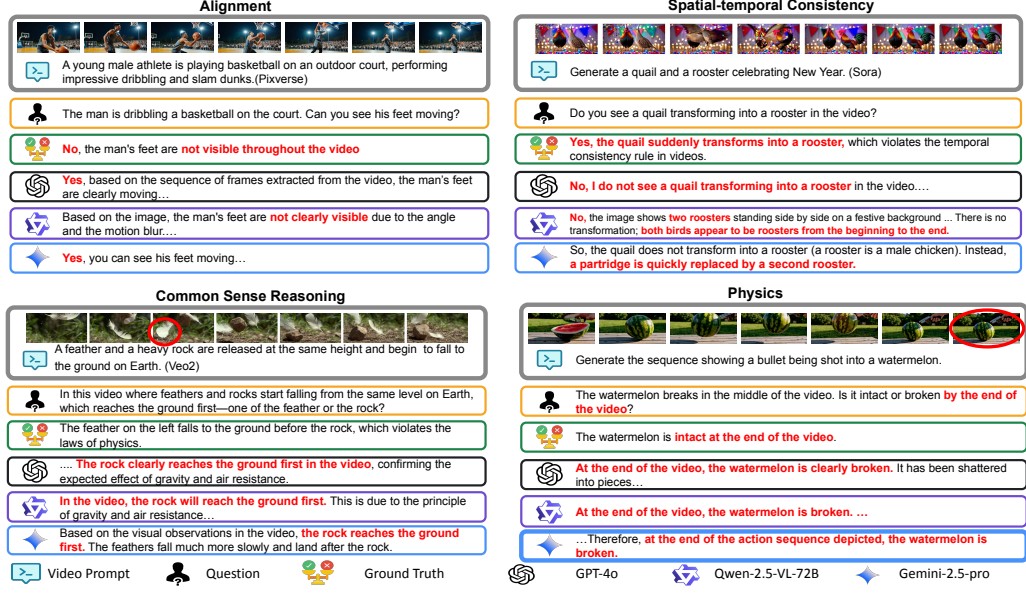

Figure 3: **Example Synthetic Videos in VideoHallu.** Example hallucination cases observed during SOTA VLM evaluations on synthetic video tasks. Each example includes the generation prompt, key frames, questions, human-annotated ground truth, and hallucinated answers from GPT-4o, Qwen2.5-VL, and Gemini-2.5-Pro, with hallucinations marked in **Red**.

## 3.1   Limitations of VLMs in OOD Data

**VLMs struggle with counterintuitive phenomena and abnormalities in generated videos.**    State-of-the-art VLMs achieve below 55% accuracy on our synthetic video QA dataset, only slightly above the 50% random baseline (Table 1). They particularly falter in commonsense and physical reasoning, often failing to detect abnormalities or relying on linguistic shortcuts instead of visual evidence. As shown in Figure 3, none of the models recognize the implausibility of a shattered watermelon reassembling, nor do they notice abrupt, counterfactual entity changes, such as a quail suddenly turning into a rooster. These failures highlight VLMs' limited capacity for abnormality reasoning and critical visual understanding beyond text priors.

**Chain-of-thought reasoning learned from real-world videos provides limited benefit for understanding synthetic videos.** VLMs trained with reinforcement learning, such as GRPO [37] (R1-finetuned) used in the DeepSeek series [21], show potential for improving reasoning and critical thinking abilities in reasoning-heavy tasks like mathematics, real-world video understanding [38, 3, 39]. This raises a question: does RL truly improve visual reasoning in VLMs, or does it only optimize for correct answers without enhancing actual visual understanding? Table 1 evaluates two R1-finetuned models, Video-R1 [7] and VideoChat-R1 [8], using chain-of-thought prompts. Both models show limited improvement compared to their base models (Qwen2.5-VL-7B) on synthetic video understanding, with minimal or worse alignment and commonsense, suggesting that training on real-world videos only inculcates real-world reasoning patterns.

**Solely pre-training on real-world data biases visual grounding.** While RL improves reasoning on math and real-world videos, it does not help with counterintuitive synthetic content that contradicts real-world norms and cannot elicit video-grounded critical thinking. In such cases, chain-of-thought prompting can bias the LLM backbone to rely too heavily on prior commonsense knowledge, neglecting synthetic visual cues and leading to hallucinated responses [9]. For instance, in the third case in Figure 3, the video shows the feather dropping to the ground before the rock when falling from the same height. When asked which reaches the ground first—the feather or the rock, VideoChat-R1-think responds: *"The video shows a feather and a rock being dropped...This is a classic demonstration of Galileo's principle; therefore the rock drops before the feather..."* While this language explanation alone is grounded in correct physical principles, it directly contradicts what actually occurs in the video. The model generates an incorrect conclusion based on prior language reasoning, showing how chain-of-thought prompting can amplify reliance on language priors and

increase hallucination risk when understanding synthetic videos that do not align with real-world expectations.

## 3.2   Visual Learning or Pattern Matching? Questioning RL's True Impact on Vision Models

Current VLMs struggle with counter-intuitive questions and critical visual thinking in synthetic videos (Section 3.1). They frequently hallucinate and a dearth of critical thinking leads them to overlook abnormality examples in VideoHallu, relying on the model's language knowledge instead of reasoning directly from the visual input. This raises a crucial question: *Can VLMs learn counter-intuitive commonsense knowledge and improve their critical thinking abilities for detecting abnormalities through training with synthetic video data?*

While supervised fine-tuning (SFT) or RL from human feedback (RLHF) are natural approaches for improving VLMs, the distributional gap between standard alignment-based video QA tasks used in pre-training and the specialized critical reasoning required for synthetic videos poses an impediment. Since synthetic videos are scarce in typical pre-training corpora, models lack sufficient exposure to develop robust reasoning capabilities for such content.

To investigate whether incorporating synthetic data alongside real data can improve VLM performance on synthetic videos, we pose two primary research questions:

- *Between SFT and GRPO training, which approach more effectively enables VLMs to develop a genuine understanding of synthetic videos?*
- *Is synthetic data in the training mixture necessary for improving model reasoning abilities on synthetic videos, or can training on real data alone suffice?*

**Method Overview: SFT vs. GRPO.**   We compare two training paradigms—SFT and GRPO [21]—to compare their effectiveness and generalization.

**Supervised Fine-Tuning (SFT).** SFT directly optimizes the model to predict ground-truth responses using maximum likelihood estimation. For a dataset $\mathcal{D} = \{(x_i, y_i)\}_{i=1}^{N}$ containing video-question pairs $x_i$ and corresponding answers $y_i$, the SFT objective minimizes the negative log-likelihood:

$$\mathcal{L}_{\text{SFT}}(\theta) = -\frac{1}{N} \sum_{i=1}^{N} \sum_{t=1}^{|y_i|} \log p_\theta \left( y_i^{(t)} \mid x_i, y_i^{(<t)} \right) \tag{1}$$

where $\theta$ represents model parameters, $y_i^{(t)}$ is the $t$-th token in sequence $y_i$, and $y_i^{(<t)}$ denotes all preceding tokens.

**Group Relative Policy Optimization (GRPO).** GRPO, a variant of reinforcement learning from human feedback, optimizes the model using preference-based learning without requiring explicit reward models. Given preference pairs $(y_w, y_l)$ where $y_w$ is preferred over $y_l$ for prompt $x$, GRPO maximizes the likelihood of preferred responses while penalizing less preferred ones:

$$\mathcal{L}_{\text{GRPO}}(\theta) = -\mathbb{E}_{(x,y_w,y_l)\sim\mathcal{D}} \left[ \log \sigma \left( \beta \log \frac{p_\theta(y_w \mid x)}{p_{\text{ref}}(y_w \mid x)} - \beta \log \frac{p_\theta(y_l \mid x)}{p_{\text{ref}}(y_l \mid x)} \right) \right] \tag{2}$$

where $p_{\text{ref}}$ is a reference model (typically the pre-trained checkpoint), $\beta$ is a temperature parameter controlling the strength of the KL penalty, and $\sigma$ is the sigmoid function. This approach encourages the model to generate responses that align better with the training data distribution while maintaining proximity to the reference policy.

The key distinction lies in their learning signals: SFT learns from direct supervision with ground-truth labels, while GRPO learns from comparative preferences, potentially enabling more internal reasoning from the model for video understanding.

**Experimental Setup and Results.** Both research questions require training data. We combine our 800 synthetic video training data with 2,000 video QA pairs sampled from Video-R1 training data derived from LLaVA-Video [40].

To keep a fair comparison across different finetuning methods while reducing the training resources needed, we use 15 frames during training with learning rate $1e^{-6}$ to train the model for one epoch using the Open-R1 [38] framework on eight A100 80G GPUs. For GRPO training, since our answers are free-form answers, we use the average ROUGE-1, ROUGE-2, ROUGE-L score [41] as the reward:

$$\text{Reward}(a_{\text{pred}}, a_{\text{gold}}) = \frac{1}{3} \sum_{i \in \{1,2,L\}} \text{ROUGE-}i(a_{\text{pred}}, a_{\text{gold}}), \qquad (3)$$

where ROUGE captures $n$-gram overlap F-score between expected answers and generated responses.

**Result: SFT VS. GRPO.** To address our first research question, we use both SFT and GRPO to train models on the mixed dataset and evaluate their performance on out-of-distribution synthetic video understanding. To validate the generalization of our findings, we run experiments on two architectures: Qwen2.5-VL-7B and LLaVA-One-Vision [32].

GRPO outperforms SFT on out-of-distribution and critical visual understanding tasks (Table 2), consistent with Feng et al. [42], who demonstrated GRPO's superior generalization. Because our dataset contains genuinely out-of-distribution synthetic videos generated by diffusion models unseen during pre-training, these results offer stronger evidence of the two paradigms' differences. GRPO's advantage indicates that SFT tends to memorize surface-level patterns, whereas GRPO cultivates reasoning skills that better transfer to novel scenarios, a key capability for synthetic video understanding, where visual and temporal dynamics diverge markedly from natural videos.

| Model | Alignment | Physics | Consistency | Commonsense | Overall |
|---|---|---|---|---|---|
| *Previous Base Models* | | | | | |
| LLaVA-OneVision [32] | 44.22 | 32.46 | 32.13 | 45.00 | 39.93 |
| Qwen2.5-VL-7B [20] | 58.02 | 44.59 | 46.99 | 47.50 | 52.98 |
| Video-R1 [7] | 58.14 | 48.20 | 49.00 | 38.75 | 53.64 |
| *SFT vs. GRPO* | | | | | |
| Qwen2.5-VL-7B SFT | 55.22 | 45.90 | 47.39 | 35.00 | 51.02 |
| Qwen2.5-VL-7B GRPO | **62.18** | **53.77** | **56.63** | 45.00 | **57.69** |
| LLaVA-OneVision SFT | 44.67 | 26.23 | 33.33 | 38.75 | 38.82 |
| LLaVA-OneVision GRPO | 46.24 | 30.82 | 34.54 | 48.75 | 41.38 |
| *Real Data vs. Synthetic Data (GRPO)* | | | | | |
| Qwen2.5-VL-7B Real Only | 57.35 | 46.89 | 51.41 | 33.75 | 53.05 |
| Qwen2.5-VL-7B Synthetic Only | 60.16 | 48.20 | 48.19 | **52.50** | 55.41 |
| Qwen2.5-VL-7B Combined | **62.18** | **53.77** | **56.63** | 45.00 | **57.69** |

Table 2: **Fine-tuning results for SFT and GRPO.** GRPO training leads to better improvement than SFT; augmenting the small synthetic video data leads to higher accuracy than training on just real videos or limited synthetic videos.

**Result: Effect of Training with Synthetic Videos.** To address our second research question regarding the relative contributions of real-world and synthetic video data to GRPO training performance, ablate the training for models via three different data configurations: (1) a combined dataset mixing both data types, (2) synthetic videos only, and (3) real-world videos only. Training only on real-world videos leads to minimal improvement (0.07%) over the base model on synthetic video understanding tasks (Table 2): real-world video training alone does not transfer effectively to the reasoning required for synthetic video analysis. In contrast, synthetic videos improve the model's detection of abnormalities in synthetic content. However, the limited size of our synthetic video training set necessitates data augmentation: combining real-world and synthetic videos in the training mixture produces the most effective results. The mixed dataset setting enables VLMs to better adapt their reasoning capabilities to synthetic videos, outperforming both single-domain training approaches. While synthetic data is crucial for developing domain-specific reasoning skills, the additional diversity provided by real-world videos helps stabilize training and improve overall robustness. Thus, VLMs require exposure to synthetic video data during training to develop effective reasoning abilities for synthetic content, and a balanced mixture of real and synthetic data optimizes out-of-distribution synthetic video understanding tasks.

**Result: Performance on Real-world Benchmark.** Incorporating synthetic video data alongside real-world videos can improve VLMs' understanding of synthetic videos. But does synthetic video training come at the cost of degraded real-world video comprehension?

To investigate this, we evaluate our trained Qwen models on two real-world benchmark datasets: MVBench [43], a comprehensive benchmark for evaluating temporal understanding and reasoning in videos, and MMVU [44], which tests expert-level multidisciplinary video understanding across diverse domains. Synthetic and real-world video understanding abilities can coexist.

| Model | MMVU | MVBench |
|---|---|---|
| Qwen-2.5VL-7B (base) | 58.7 | 69.6 |
| + Real Only | 61.3 | 70.9 |
| + Synthetic Only | 60.1 | 70.1 |
| + Combined | 61.3 | 70.1 |

Table 3: Post-training performance on real-world video understanding benchmarks.

**Discussion.** Throughout the evaluations over our benchmark and the fine-tuning over pre-trained VLMs, we gather essential insights to accelerate further improvement over future VLMs for synthetic video understanding. We list them as follows:

**1. VLMs hallucinate on Synthetic Data due to Neglect of Actual Visual Content.** As shown in Table 1 and Figure 17, all tested SOTA VLMs, including large models like Qwen2.5-VL (7B/32B), GPT-4o, and Gemini-2.5-Pro, as well as smaller models (<7B), struggle with counterintuitive QA on synthetic videos in VideoHallu. One reason is that VLMs often solely rely on their embedded commonsense and physics priors to answer questions, even when prompted to rely on video content (Figure 3). These hallucinations, caused by misalignment between video context and real-world norms, are rare in real-world QA but prevalent in synthetic settings, particularly for counterfactual reasoning. Although VLMs are exposed to some synthetic data during training, the vast majority of their training consists of real-world videos that follow physical laws and commonsense principles. Consequently, VLMs treat such rules as universal priors that override visual evidence, leading to hallucinations when synthetic videos contradict learned physical principles.

**2. Critical thinking may be biased by language priors in synthetic visual abnormality detection.** As discussed in Section 3.1, while RL enhances critical thinking in real-world video QA, all R1-trained VLMs we evaluated, such as Video-R1-CoT and VideoChat-R1-think in Table 1, consistently underperform their base model (Qwen2.5-VL-7B) on VideoHallu, showing no clear improvement on commonsense or physics-oriented questions. We attribute this to flawed reasoning patterns in R1-trained VLMs. Although chain-of-thought reasoning can elicit more structured inference in real-world contexts, it proves ineffective in synthetic video settings, where detecting abnormalities demands grounded, fine-grained visual understanding. R1-trained models often excel in language-only reasoning tasks [21, 45, 37], yet when extended to multimodal domains, their reasoning becomes heavily rely on linguistic priors. Consequently, their CoT responses tend to reflect superficial comprehension of visual content and are more susceptible to hallucinations in counterintuitive or visually deceptive scenarios [9, 46].

**3. The high-quality negative control examples matter for model improvement.** Given the need to improve VLMs' performance in synthetic video abnormality detection, as shown in Section 3.1, we run RL training experiments over Qwen2.5-VL and LLaVa-One-Vision with a mixture of real-world and synthetic videos. Our results show that, after training models with some synthetic videos, VLMs show improvements in critical thinking and their ability to handle counterintuitive scenarios. Our results suggest that it is the quality and coverage of the data, not just the fine-tuning method, that drive gains. With a small but well-annotated dataset containing both positive and negative examples, detailed reasoning steps, and reasoning-oriented training like GRPO, even small models like Qwen2.5-VL-7B show improved QA accuracy. This highlights the importance of high-quality, reasoning-rich data in helping VLMs internalize and apply commonsense and physics knowledge, even with limited post-training resources.

## 4 Related Work

**Hallucinations in VLMs.** Hallucinations refer to the persistent challenge of generating outputs that contradict or misrepresent the target texts, images, or videos [47, 48]. It arises from conflicts between the language priors of VLMs and the actual visual inputs [49], which is more severe in

video understanding than in static image understanding due to the complex entanglement of spatial-temporal information across the timeline and the contextual cues associated with entities within frames. A line of prior work, such as VideoHallucer [50], EventHallusion [51], and HAVEN [52], established benchmarks for evaluating model hallucination on both entities and events within videos, while also proposing methods to enhance the video understanding capabilities of VLMs [4, 53, 54]. However, most prior works on hallucination, particularly in the video domain, rely on real-world factual data, rather than synthetic data generated by generative models. Hallucination in generative video understanding models remains an open and largely unexplored research area.

**Reinforcement Learning for Post-training of Vision-Language Models.** Inspired by the techniques from DeepSeek-R1 [21], there is an increasing body of research that leverages reinforcement learning in post-training to enhance the general-purpose multimodal reasoning capabilities of VLMs [55, 56]. Most recent efforts have focused on using GRPO and its variants to fine-tune VLMs to elicit more robust reasoning and perception skills [57, 56, 3, 58]. The representative work Video-R1 [7] collects a large-scale corpus of 260K video and image samples and performs GRPO with data type–specific reward engineering. It applies regression-based approximations for numeric answer types, ROUGE-based metrics for free-form textual responses, and exact match rewards for multiple-choice questions, enhancing models' temporal reasoning for real-world video understanding. VideoChat-R1 [59] extends this paradigm to interactive multimodal dialogue, combining video-centric instruction tuning with reinforcement learning from human feedback (RLHF). Nonetheless, prior research has predominantly focus on visual understanding in real-world imagery and videos, with generated videos receiving comparatively little attention.

**Video Generation Models and Synthetic Content Monitoring.** Recent advances in video generation models have enabled the creation of highly realistic and aesthetic videos from text prompts, reference images, or conditioning frames [14, 23, 12, 25, 13, 26]. These models are increasingly applied in content creation, simulation for robotics and autonomous driving [60]. Since the release of Veo3 [14], Sora 2 [61], Wan [62], the volume of generated content has exploded. These vast generated videos present major challenges for content monitoring, quality evaluation, and content verification. Manual annotation and evaluation are increasingly infeasible given the scale and variability of generated outputs, thus motivating the need for automated, scalable evaluation and reasoning frameworks that are specifically tuned for synthetic video understanding. To date, a few works have explored using VLMs to evaluate generated images (for example, detection of synthetic images or assessing image generation quality). For example, [63] presents a method named Bi-LORA that uses a VLM to detect synthetic images. However, the domain of synthetic videos is still largely under-explored: we lack systematic methods, datasets and evaluation protocols for using VLMs to judge and understand synthetic videos

# 5 Conclusion and Limitation

**Conclusion.** We introduce VideoHallu, a dataset designed to evaluate VLMs' visual common-sense and physics reasoning through synthetic videos with counterfactual scenarios. It features expert-annotated, reasoning-intensive QA pairs spanning alignment, spatial-temporal consistency, commonsense, and physics categories to assess VLMs' ability to detect abnormalities and violations of physical laws. Evaluation of SOTA VLMs on VideoHallu shows hallucinations and critical thinking failures. Fine-tuning with GRPO with both real and synthetic videos leads to accuracy improvements on VideoHallu, showing the value of incorporating structured physics and commonsense reasoning data to improve VLM performance on synthetic video tasks. However, scalability remains a limitation, as generating high-quality annotations and fine-tuning VLMs at scale is costly, and limited access to data and compute constrains further progress. Future work will focus on expanding synthetic video datasets with abnormality QA pairs to train VLMs for critical, visually-grounded reasoning. Scaling with adversarial QA pairs can enhance robustness and enable automatic video evaluation via prompt decomposition, reducing reliance on human annotations.

**Limitations.** Despite enabling controlled evaluation of visual commonsense and physics reasoning, VideoHallu has three key limitations: (i) a *domain gap* between synthetic videos and real-world visual complexity, which may reduce transfer; (ii) limited *coverage* of rare physical edge cases and long-horizon causal interactions; and (iii) high *scalability cost* for expert annotation, counterfactual generation, and GRPO fine-tuning, constrained by data and compute access. Moreover, QA-based

scoring may not fully reflect grounding or uncertainty calibration, motivating complementary metrics and automated adversarial QA generation.

## 6   Acknowledgment

The work was done with the computer resources and support of Lambda Lab. Boyd-Graber and Li are supported by National Science Foundation Grant No. IIS-2403436. Any opinions, findings, and conclusions or recommendations expressed in this material are those of the author(s) and do not necessarily reflect the views of the National Science Foundation.

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

# A   More Synthetic Video Examples

We present selected cases from SOTA MLLM evaluations across each VideoHallu sub-category. Hallucinations in model answers, common sense or physics violations in videos, and other notable cues in the video, questions, or ground truth are highlighted to assist the reader's understanding.

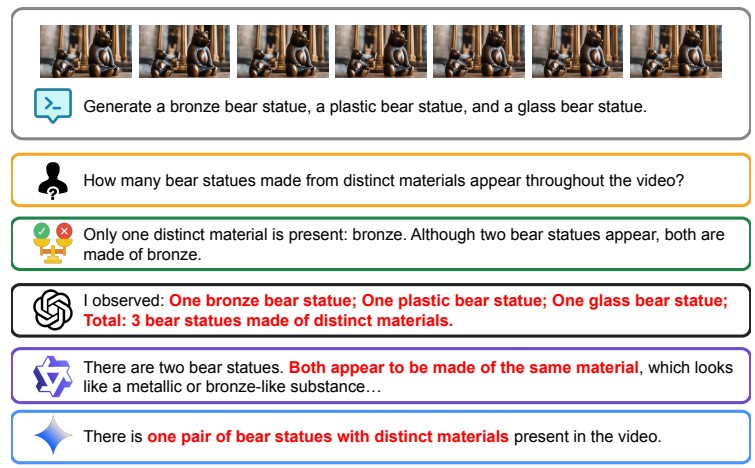

Alignment - Entity Counting

Figure 4: **Hallucination Case from Alignment – Entity Counting (A-EC).** We show hallucination examples from SOTA MLLM evaluations under the A-EC category. Each case includes the video generation prompt (**Gray**), key frames from synthetic videos (**Gray**), questions (**Orange**), ground truth (**Green**), and model answers from GPT-4o (**Black**), Qwen2.5-VL (**Purple**), and Gemini-2.5-Pro (**Blue**), with hallucinations and critical context highlighted in **Red**.

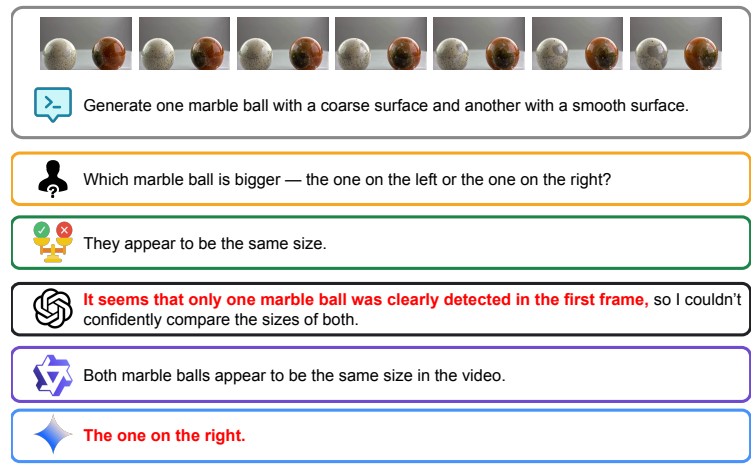

Alignment - Entity Properties

Figure 5: **Hallucination Case from Alignment – Entity Properties (A-EP).** We show hallucination examples from SOTA MLLM evaluations under the A-EP category. Each case includes the video generation prompt (**Gray**), key frames from synthetic videos (**Gray**), questions (**Orange**), ground truth (**Green**), and model answers from GPT-4o (**Black**), Qwen2.5-VL (**Purple**), and Gemini-2.5-Pro (**Blue**), with hallucinations and critical context highlighted in **Red**.

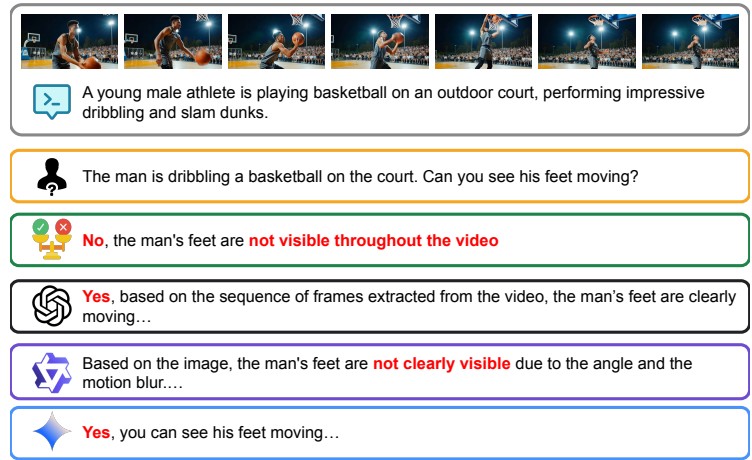

Alignment - Entity Recognition and Classification

Figure 6: **Hallucination Case from Alignment – Entity Recognition and Classification (A-ERAC).** We show hallucination examples from SOTA MLLM evaluations under the A-ERAC category. Each case includes the video generation prompt (**Gray**) key frames from synthetic videos (**Gray**), questions (**Orange**), ground truth (**Green**), and model answers from GPT-4o (**Black**), Qwen2.5-VL (**Purple**), and Gemini-2.5-Pro (**Blue**), with hallucinations and critical context highlighted in **Red**.

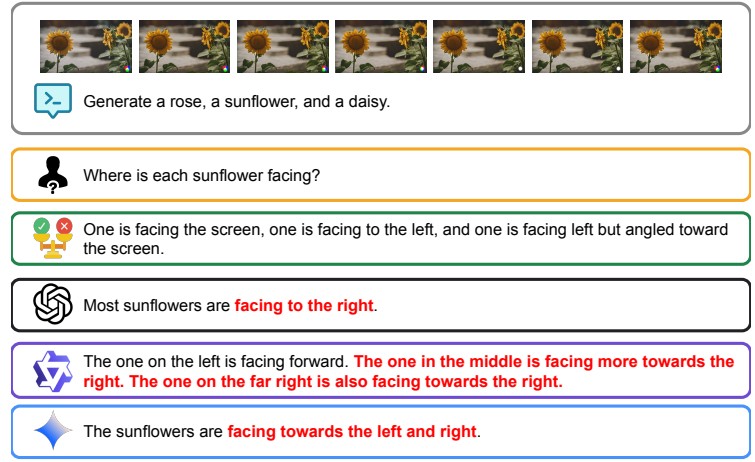

Alignment - Spatial Relationships

Figure 7: **Hallucination Case from Alignment – Spatial Relationships (A-SR).** We show hallucination examples from SOTA MLLM evaluations under the A-SR category. Each case includes the video generation prompt (**Gray**), key frames from synthetic videos (**Gray**), questions (**Orange**), ground truth (**Green**), and model answers from GPT-4o (**Black**), Qwen2.5-VL (**Purple**), and Gemini-2.5-Pro (**Blue**), with hallucinations and critical context highlighted in **Red**.

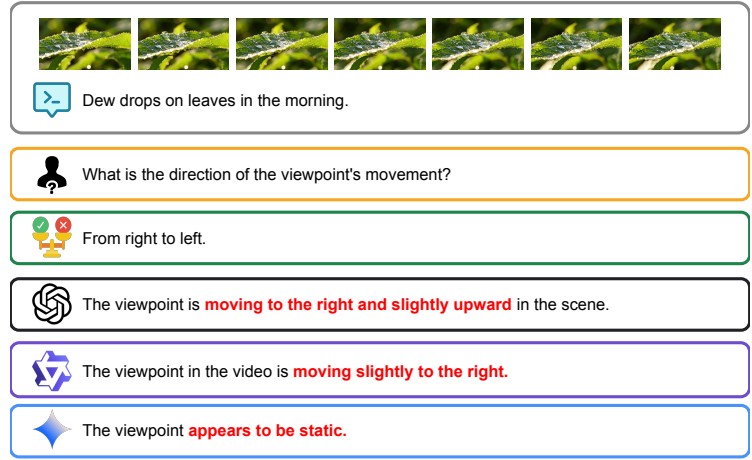

Spatial-temporal Consistency - Camera Dynamics

Figure 8: **Hallucination Case from Spatial-temporal Consistency – Camera Dynamics (SC-CD).** We show hallucination examples from SOTA MLLM evaluations under the SC-TD category. Each case includes the video generation prompt (**Gray**), key frames from synthetic videos (**Gray**), questions (**Orange**), ground truth (**Green**), and model answers from GPT-4o (**Black**), Qwen2.5-VL (**Purple**), and Gemini-2.5-Pro (**Blue**), with hallucinations and critical context highlighted in **Red**.

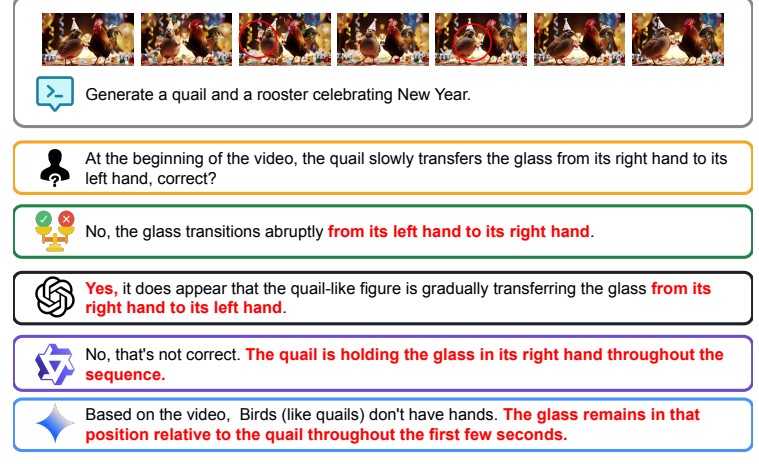

Spatial-temporal Consistency - Spatial Dynamics

Figure 9: **Hallucination Case from Spatial-temporal Consistency – Spatial Dynamics (SC-SD).** We show hallucination examples from SOTA MLLM evaluations under the SC-SD category. Each case includes the video generation prompt (**Gray**), key frames from synthetic videos (**Gray**), questions (**Orange**), ground truth (**Green**), and model answers from GPT-4o (**Black**), Qwen2.5-VL (**Purple**), and Gemini-2.5-Pro (**Blue**), with hallucinations and critical context highlighted in **Red**.

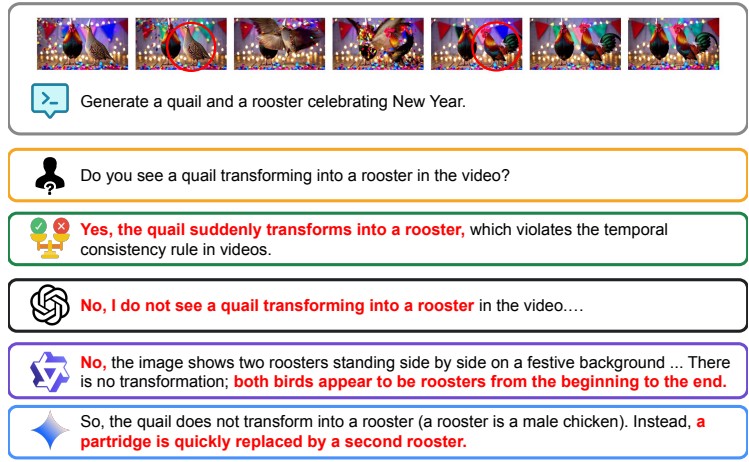

Spatial-temporal Consistency - Temporal Dynamics

Figure 10: **Hallucination Case from Spatial-temporal Consistency – Temporal Dynamics (SC-TD).** We show hallucination examples from SOTA MLLM evaluations under the SC-TD category. Each case includes the video generation prompt (Gray), key frames from synthetic videos (Gray), questions (Orange), ground truth (Green), and model answers from GPT-4o (Black), Qwen2.5-VL (Purple), and Gemini-2.5-Pro (Blue), with hallucinations and critical context highlighted in Red.

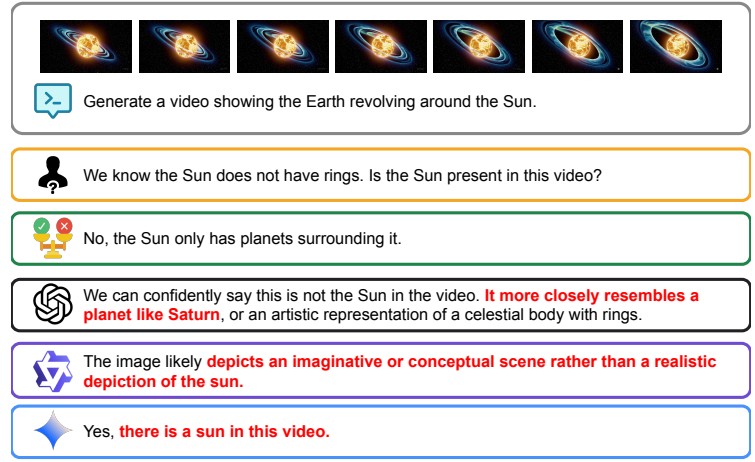

Common Sense Reasoning - Knowledge

Figure 11: **Hallucination Case from Common Sense Reasoning - Knowledge (CS-K).** We show hallucination examples from SOTA MLLM evaluations under the CS-K category. Each case includes the video generation prompt (Gray), key frames from synthetic videos (Gray), questions (Orange), ground truth (Green), and model answers from GPT-4o (Black), Qwen2.5-VL (Purple), and Gemini-2.5-Pro (Blue), with hallucinations and critical context highlighted in Red.

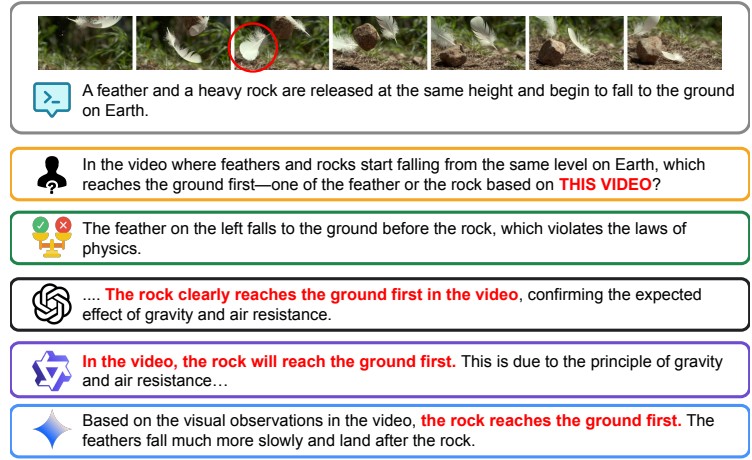

Common Sense Reasoning - Reasoning

Figure 12: **Hallucination Case from Common Sense Reasoning - Reasoning (CS-R).** We show hallucination examples from SOTA MLLM evaluations under the CS-R category. Each case includes the video generation prompt (Gray), key frames from synthetic videos (Gray), questions (Orange), ground truth (Green), and model answers from GPT-4o (Black), Qwen2.5-VL (Purple), and Gemini-2.5-Pro (Blue), with hallucinations and critical context highlighted in Red.

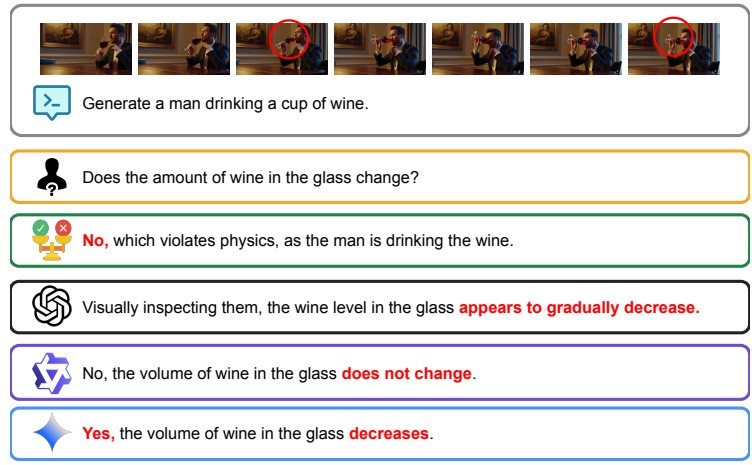

Physics - Conservation

Figure 13: **Hallucination Case from Physics - Conservation (P-C).** We show hallucination examples from SOTA MLLM evaluations under the P-C category. Each case includes the video generation prompt (Gray), key frames from synthetic videos (Gray), questions (Orange), ground truth (Green), and model answers from GPT-4o (Black), Qwen2.5-VL (Purple), and Gemini-2.5-Pro (Blue), with hallucinations and critical context highlighted in Red.

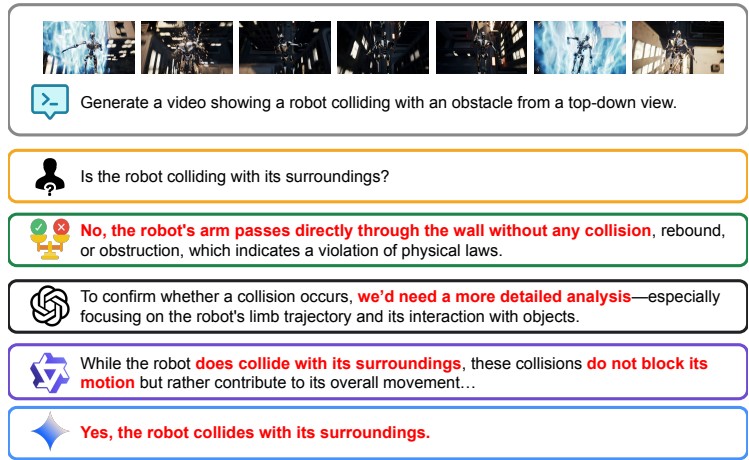

Physics - Constraints and Properties

Figure 14: **Hallucination Case from Physics - Constraints and Properties (P-CAP).** We show hallucination examples from SOTA MLLM evaluations under the P-CAP category. Each case includes the video generation prompt (Gray), key frames from synthetic videos (Gray), questions (Orange), ground truth (Green), and model answers from GPT-4o (Black), Qwen2.5-VL (Purple), and Gemini-2.5-Pro (Blue), with hallucinations and critical context highlighted in Red.

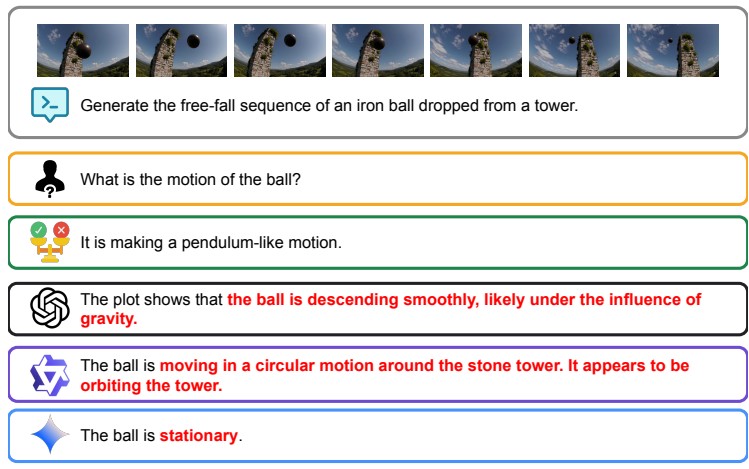

Physics - Motion

Figure 15: **Hallucination Case from Physics - Motion (P-M).** We show hallucination examples from SOTA MLLM evaluations under the P-M category. Each case includes the video generation prompt (Gray), key frames from synthetic videos (Gray), questions (Orange), ground truth (Green), and model answers from GPT-4o (Black), Qwen2.5-VL (Purple), and Gemini-2.5-Pro (Blue), with hallucinations and critical context highlighted in Red.

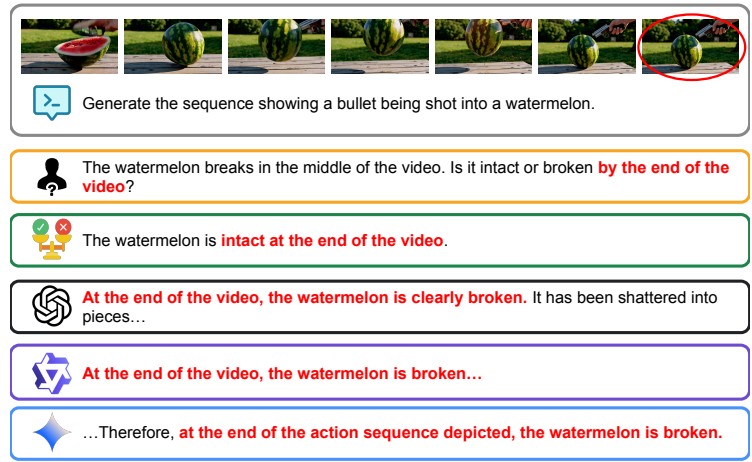

Generate the sequence showing a bullet being shot into a watermelon.

The watermelon breaks in the middle of the video. Is it intact or broken **by the end of the video**?

The watermelon is **intact at the end of the video**.

**At the end of the video, the watermelon is clearly broken.** It has been shattered into pieces…

**At the end of the video, the watermelon is broken…**

…Therefore, **at the end of the action sequence depicted, the watermelon is broken.**

Physics - State Transition

Figure 16: **Hallucination Case from Physics - State Transition (P-ST).** We show hallucination examples from SOTA MLLM evaluations under the P-ST category. Each case includes the video generation prompt (**Gray**), key frames from synthetic videos (**Gray**), questions (**Orange**), ground truth (**Green**), and model answers from GPT-4o (**Black**), Qwen2.5-VL (**Purple**), and Gemini-2.5-Pro (**Blue**), with hallucinations and critical context highlighted in **Red**.

# B Theoretical Problem Formulation

The motivation of our work comes from the assumption that the language priors within the LLM backbone of the VLMs may interfere with their understanding of synthetic videos. Our goal is to craft a dataset of synthetic videos featuring perceptually obvious violations of common sense and physical laws that require true visual recognition to detect. Let $f_{\text{VLM}}$, $f_{\text{LLM}}$ denote the VLM and its LLM backbone, respectively, and $f_{\text{Human}}$ denote the human expert providing ground truth understanding. $f_{\text{VLM}}(video, query)$ can take a video-query pair as input, $f_{\text{LLM}}(context, query)$ can take a text-only context-query pair as input, and $f_{\text{Human}}(context, query)$ can take multi-modal inputs paired with queries. We denote $\mathcal{V}$ as the set of all contexts within the synthetic video $V$. The context $\mathcal{C}$ denotes the context being probed during the video understanding process, where $\mathcal{Q}$ denotes the query probing this context $\mathcal{C}$. We define a mapping function $T(\cdot)$ that transforms a set of contextual elements into a natural language-formulated text for both the query $\mathcal{Q}$ and context $\mathcal{C}$. This mapping can be performed by either humans or LLMs. We introduce the *contextual distance* $d[\cdot, \cdot]$ to quantify the semantic divergence between two contexts or texts [49]. When two contexts convey similar or mutually consistent information, $d$ is small; otherwise, it is large. This metric captures the degree of contextual alignment and can be estimated using *LLM-as-a-Judge* approaches [27–29] or other model-based evaluators. In the post-training *human preference alignment* setting, we regard $f_{\text{Human}}(\cdot, \cdot)$ as the ground truth and expect both $f_{\text{VLM}}$ and $f_{\text{LLM}}$ to align with human perception and understanding of the real world. The objective is formulated as:

$$\max_{V,\mathcal{Q},\mathcal{C}} \quad d[f_{\text{VLM}}(V, T(\mathcal{Q})), f_{\text{Human}}(V, T(\mathcal{Q}))] \tag{4}$$

$$\text{s.t.} \quad d[f_{\text{LLM}}(T(\mathcal{C}), T(\mathcal{Q})), f_{\text{Human}}(T(\mathcal{C}), T(\mathcal{Q}))] \leq \epsilon,$$

$$d[f_{\text{Human}}(T(\mathcal{C}), T(\mathcal{Q})), f_{\text{Human}}(V, T(\mathcal{Q}))] \geq \delta, \ \mathcal{C} \subseteq \mathcal{V}, \tag{5}$$

where Equation (4) maximizes the contextual distance between the VLM's output and the human-annotated ground truth for a given synthetic video $V$ and query $\mathcal{Q}$. The constraints in (5) ensure that the language-only model $f_{\text{LLM}}$, given the same query $\mathcal{Q}$ and context $\mathcal{C}$, remains aligned with human judgment within a tolerance $\epsilon$, while the video $V$ introduces human-detectable inconsistencies relative to $\mathcal{C}$, yielding a contextual distance exceeding a threshold $\delta$. The context $\mathcal{C}$ is embedded within $V$ to preserve coherence.

# C   Video Understanding and Evaluation Categorization/Motivation

We provide details on specific categorizations of errors video generation models can make. We draw inspiration from basic video quality evaluation definitions from MVBench [43] and WorldModel-Bench [64] to first organize the current challenges of video generations and evaluations in four basic categories (Figure 2). Given the probing target of each question-answering pair and the demand for reasoning abilities or prior knowledge of the LLM backbone to solve the question provided, we divide the question-answering pairs for testing MLLM-as-evaluators into four major categories with sub-categories.

The categorization is to go beyond superficial metrics like frame consistency or resolution by enabling rigorous evaluation through the identification of visual abnormalities across predefined categories. To achieve this, we design targeted adversarial questions that expose these anomalies. This allows us to assess whether current SOTA MLLMs can effectively detect and interpret such issues, which is an essential step toward scalable and interpretable video evaluation. We further extend these principles to define our video understanding criteria benchmark.

**Alignment** checks whether the model accurately identifies basic entity details and ensures the video content fully aligns with the prompt without omissions or discrepancies.

- **Entity Counting (A-EC):** Quantifies how many entities are present in the scene.
- **Entity Properties (A-EP):** Focuses on visual features such as color, shape, and texture that define an entity's appearance.
- **Entity Recognition and Classification (A-ERAC):** Identifies and categorizes entities based on attributes like shape, color, and texture.
- **Spatial Relationships (A-SR):** Examines the relative positions of mostly static entities as described in the prompt.

**Spatial-Temporal Consistency** evaluates whether the model can detect smooth, consistent changes in objects, actions, and viewpoints over time, without abrupt or abnormal transitions in space or time.

- **Camera Dynamics (SC-CD):** Covers variations in camera movement, angle, and viewpoint.
- **Spatial Dynamics (SC-SD):** Focuses on entity motion, changing positions, and interactions, identifying any inconsistencies or abrupt spatial changes.
- **Temporal Dynamics (SC-TD):** Tracks changes in entities or scenes over time, including appearance shifts, transformations, and abnormal appearances or disappearances.

**Common Sense Reasoning** assesses the model's ability to apply general knowledge and reasoning to detect conflicts between common sense and the visual context, ensuring it interprets the prompt correctly without hallucinating entities or actions.

- **Knowledge (CS-K):** Assesses the model's ability to apply general knowledge of everyday phenomena, including object geometry, layout, and state transitions.
- **Reasoning (CS-R):** Tests the model's ability to interpret problem cues—including emotional or environmental hints, and solve them through reflection and chain-of-thought.

**Physics** assesses the model's ability to detect physical inconsistencies, such as violations of gravity, motion dynamics, or conservation laws, requiring careful reasoning about object properties and movements even if not explicitly stated.

- **Conservation (P-C):** Assesses understanding of mass and energy conservation, ensuring entity quantities remain constant unless acted upon by external forces.
- **Constraints and Properties (P-CAP):** Checks understanding of physical constraints and properties, such as rigid bodies blocking motion or light behavior like reflection.
- **Motion (P-M):** Evaluates the model's grasp of motion-related physics (like gravity, linear/circular motion, relative movement, and fluid dynamics), spotting inconsistencies or abrupt changes.
- **State Transition (P-ST):** Tests knowledge of physics-driven state changes, including heat effects, phase transitions, and dynamic interactions.

# D  Prompt Templates

We provide the prompt templates we use for CoT prompt (Table 4) then generate the final answer (Video-R1-CoT and VideoChat-R1-thinking) and prompt templates for generating answers directly (Table 5).

---

**CoT Prompt Template**

---

System Prompt:  A conversation between User and Assistant.  The user asks a question, and the Assistant solves it.  The assistant first thinks about the reasoning process in the mind and then provides the user with the answer.  The reasoning process and answer are enclosed within <think> </think> and <answer> </answer> tags, respectively, i.e., <think> reasoning process here </think><answer> answer here </answer>

Input:  Please think about this question as if you were a human pondering deeply.  Engage in an internal dialogue using expressions such as 'let me think', 'wait', 'Hmm', 'oh, I see', 'let's break it down', etc, or other natural language thought expressions.  It is encouraged to include self-reflection or verification in the reasoning process.  Provide your detailed reasoning between the <think> </think> tags, and then give your final answer between the <answer> </answer> tags.

Question:  {Question}

---

Table 4: The prompt template for Video-R1-CoT and VideoChat-R1-thinking to generate answers. This prompt encourages them to first think critically about the video and the question then generate a final answer.

---

**Direct Answer Prompt Template**

---

System Prompt:  A conversation between User and Assistant.  The user asks a question, and the Assistant solves it.  The assistant provide answers within the <answer> </answer> tags:  <answer> answer here </answer>

Input:  You will be given a video and a question.  Please provide an answer to the question based on the video enclosed by <answer> your answer </answer> tags.

Question:  {Question}

Answer:

---

Table 5: Direct answer directly prompts a model to generate the answer without generating additional chain-of-thoughts.

---

**LLM-as-A-Judge Prompt Template**

---

You will be given a question, a reference answer, and a predicted response.  You task is to judge the correctness of the predicted response.  If the predicted response is correct, please answer "correct".  If the predicted response is incorrect, please answer "incorrect".  Please strictly follow the output format below.

OUTPUT FORMAT:

Judgment:  YOUR JUDGMENT

Question:  {Question}

Reference Answer:  {Reference Answer}

Predicted Answer:  {Predicted Response}

YOUR OUTPUT:

---

Table 6: LLM-as-a-judge prompt template.

# E Categorization Breakdown Results

We provide a qualitative breakdown of results in multiple radar charts across fine-grained categories for the evaluated baselines, serving as supplementary analysis to Table 1.

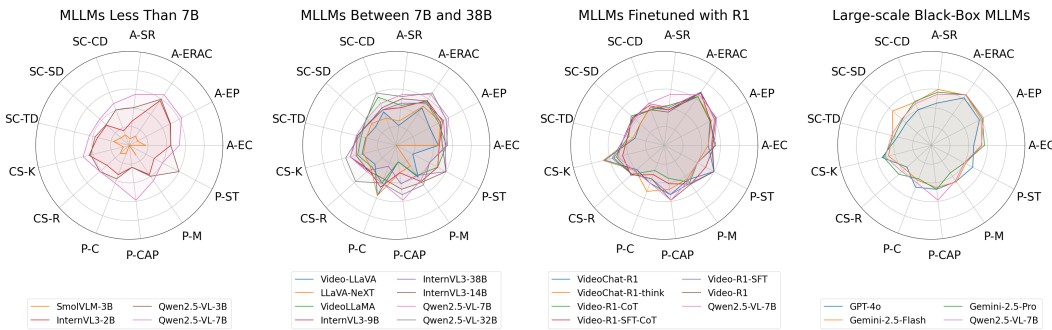

Figure 17: **SOTA VLM Evaluation on VideoHallu Across Sub-Categories.** We evaluate SOTA VLMs on VideoHallu, with results broken down by sub-category. From left to right, we show: (a) models under 7B parameters; (b) models between 7B–38B; (c) R1 fine-tuned models; and (d) large black-box VLMs. While many perform well on alignment tasks, they remain prone to hallucinations in reasoning-heavy tasks, with notably weaker performance on physics and commonsense reasoning.

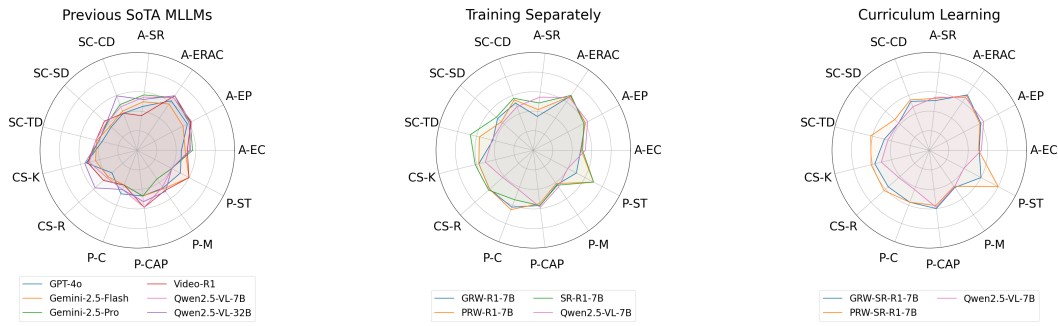

Figure 18: **Evaluation Breakdown of Fine-Tuned Models.** We show results for (a) previous SOTA VLMs, (b) models fine-tuned on sub-datasets, and (c) models fine-tuned on the full dataset via curriculum learning. Compared to the baseline (Qwen2.5-VL-7B), RFT on commonsense and physics data improves models' reasoning and overall performance in synthetic video understanding.

# F    Common Sense and Video-dependent Question-Answering

Our benchmark, VideoHallu, is designed to evaluate MLLMs' abilities to detect abnormalities in synthetic videos—a task often confounded by hallucinations stemming from commonsense or physical knowledge embedded in their language priors. This section breaks down model performance across question types in VideoHallu, including:

- **Common Sense-only Questions:** These can be answered using language priors alone, without relying on video input. *e.g., What typically happens when a bullet hits a watermelon?* (Answer: *It explodes into pieces.*)
- **Counterintuitive Questions:** Target counterfactual contexts in synthetic videos, testing whether MLLMs can recognize visually implausible phenomena. *e.g. In the video (Sora), the watermelon breaks in the middle of the video. Is it intact or broken at the end?* (Answer: *It's intact.*) (Figure 1)
- **Critical Thinking Questions:** Open-ended questions that ask whether MLLMs can identify abnormalities in synthetic videos, evaluating their visual reasoning. *e.g. What is unusual in this video (Sora)?* (Answer: *The watermelon explodes, then reassembles.*) (Figure 1)

while the latter two types of questions must be answered with video inputs, so that we denote them as video-dependent questions.

| Model | Common Sense-only | Video-dependent | | Overall |
|---|---|---|---|---|
| | | Counterintuitive | Critical Thinking | |
| GPT-4o | 100.0 | 46.8 | 15.0 | 45.5 |
| InternVL3-14B | 100.0 | 48.2 | 10.0 | 46.7 |
| Gemini-2.5-Pro | 100.0 | 50.2 | 23.3 | 49.8 |
| Video-R1 | 100.0 | 52.3 | 16.7 | 50.8 |
| Qwen2.5-VL-7B | 100.0 | 53.1 | 10.0 | 51.0 |
| Qwen2.5-VL-32B | 100.0 | 52.5 | 13.3 | 51.4 |

Table 7: **Common Sense and Video-dependent QA over VideoHallu.** We divide VideoHallu into multiple categories over the question types: **(a) Common Sense-only Questions,** answerable via language priors without video inputs; **(b) Counterintuitive Questions,** probing MLLMs' abilities in detecting counterintuitive phenomena; and **(c) Critical Thinking Questions,** assessing MLLMs' ability to detect abnormalities in synthetic videos.

In Table 7, we show the evaluation breakdown by question type for six SOTA MLLMs. All models reach 100.0% accuracy on commonsense-only questions, indicating strong grounding in pre-trained knowledge. However, performance drops on counterintuitive questions (all below 55%) and further on critical thinking questions, where no model exceeds 25% accuracy, revealing major limitations in detecting and reasoning about abnormalities based on physics and commonsense.

Gemini-2.5-Pro performs best on critical thinking (23.3%), followed by Video-R1 (16.7%), suggesting some benefit from CoT prompting. However, CoT remains unreliable under language prior bias and does not consistently improve abnormality detection. Enhancing MLLMs' critical thinking for such tasks remains an open challenge.

Counterintuitive questions typically include contextual hints, helping models locate anomalies. In contrast, critical thinking questions are open-ended, requiring models to identify and reason about abnormalities unaided, making them more vulnerable to hallucinations when their video understanding is incomplete.

