# OpenReview forum: "VideoHallu: Evaluating and Mitigating Multi-modal Hallucinations on Synthetic Video Understanding"
_NeurIPS.cc/2025/Conference — NeurIPS 2025 poster_

### Official Review · Reviewer_DNqt · 2025-06-04

**Clarity:** 2
**Significance:** 1
**Originality:** 3
**Rating:** 3
**Confidence:** 3

**Summary:**

This paper evaluates counterintuitive, commonsense, and physics reasoning in videos. It introduces a benchmark of syntetically generated videos with commonsense and physical-law violations that are perceptually clear to humans but hallucinated by MLLMs with video understanding capabilities. It then evaluates state-of-the-art MLLMs on the proposed benchmark and proposes a GRPO-based reinforcement fine-tuning to cope with this problem.

**Questions:**

**Q1:** The authors claim that: *after curriculum learning, MLLMs exhibit notable improvements in critical thinking and their ability to handle counterintuitive scenarios*, as stated based on Table 3. However, this conclusion is not sufficiently supported by the experimental evidence.
Specifically, the study evaluates RFT with curriculum learning on only a single model (Qwen2.5-VL-7B) and a single dataset (the authors' proposed benchmark).
As such, the **RTF results cannot be generalized to multiple MLLMs** without further validation. The authors should rephrase the claim or provide further experiments.

**Q2:** The authors claim that: *it is the quality and coverage of the data, not just the fine-tuning method, that drive gains* (L233-234). However, based on results in Table 3, it seems to me that Qwen2.5-VL-7B benefits from Curriculum Learning with composite datasets primarily in cases where **improvements were already evident under the Training Separately setting**. Specifically, the gains from Qwen2.5-VL-7B to the central part of the table are higer than those observed toward the end of it. Can the authors elaborate on this?

**Q3:** Implementation details are completely omitted. Key information such as the **duration of the generated videos, the number of input frames provided to MLLMs for evaluation**, and whether frame sampling is needed (and if so, which one) is entirely missing.  These are not just details but important to understand the solidity of this work.

**Q4:** The **data collection** is formulated as an optimization problem, yet the procedure seems mostly human-curated. It is not clear how the mapping T(.) is formulated. What does it mean that can be performed by either humans or LMMs? What is that exactly? Moreover, the actual hyperparameters of the constraints ($\epsilon$, $\delta$)are not stated, and it is not clear how they have been decided. Finally, what is the exact MLLM used in this step? I am not able to find this information, but I may be missing it.

**Q5:** Why did the authors not send this paper to the **Datasets and Benchmarks** track? This paper explicitly proposes a benchmarking suite (VideoHallu) and evaluates state-of-the-art models on it. Given the existence of a dedicated track with clear guidelines, including requirements for code documentation, dataset formatting, and open-sourcing by the main deadline, this raises the question of whether this submission is best suited to the current track. At a minimum, the inclusion of code and at least a sample of the dataset in the supplementary materials would have been expected. It would be helpful to understand the authors' perspective on this decision.

**Q6:** There are no information on **computing resources** used. Although in the checklist the answer to the question 8 is 'Yes', it seems that there is no information on the number and type of GPU needed for the evaluation and for the reinforcement fine-tuning. This input is needed.

**Ethical Concerns:**

["NO or VERY MINOR ethics concerns only"]

**Final Justification:**

During the rebuttal phase, the authors addressed my concerns and provided additional results using new MLLMs. However, I remain unconvinced that the benefits of curriculum learning are broadly demonstrated; improvements appear mainly in cases where gains were already observed under the ‘training separately’ setting. This issue was also raised by reviewer bcUY. Considering the new results, I will raise my score, as the current version no longer warrants a 2. Nevertheless, I believe the paper requires further development and is not yet ready for full acceptance.

**Limitations:**

The authors include limitations in the conclusions, however, this is a 2-line comment that could be further extended at least in the Appendix.

**Paper Formatting Concerns:**

I am not sure if this is against the guidelines or not, but the authors removed part of the template below the title. The 'Affiliation / Address / email' part is missing.

**Quality:**

1

**Strengths And Weaknesses:**

**Strengths:** The motivation of studying counterintuitive, commonsense, and physics reasoning in MLLMs with video understanding capabilities is interesting and well motivated. The paper evaluates several recent MLLMs, including less tested models trained with reinforcement learning or chain-of-thought supervised fine-tuning. Lastly, it proposes an approach to improve the poor performance of MLLMs on the proposed benchmark.

**Weaknesses:** While the idea is interesting, several parts of the execution need, in my opinion,  improvement. Some parts of the data collection and the conducted evaluation are not clear, as detailed in the part dedicated to questions below. Moreover, although the final section of the paper introduces a methodology aimed at enhancing MLLM capabilities, it is evaluated on a single model with results that do not convincingly support the claims made.
Both main Tables 1 and 2 are complemented with Figures that, in my opinion, are useless. Figure 3 shows exactly the same numbers in Table 2, yet without the y-axis range, so it contains even less information. The same applies to Figure 5 and Table 3. I suggest removing these figures and using the space more cleverly.

**Minor Weaknesses:**
- *Black Box* in Table 2 may be an inappropriate name. I suggest changing it to closed-source or proprietary.
- Unify LVLM and MLLM inconsistency: in the problem formulation in L87-99 LVLM is used while in the rest of the paper MLLM.
- SOTA is missing the acronym, and sometimes it is written as SoTA (L133).

---

> ### Author Rebuttal · Authors · 2025-07-31
>
> **Response to Weakness 1 :** Please see our response to the questions below.
> We run an additional different model (internVL 9B) and show the same conclusion that curriculum learning can improve MLLMs' reasoning abilities on synthetic video understanding. See Table I for reference.
>
> **Response to Weakness 2 (Figures Layout):**
> Figures 3 and 5 (radar charts) are intended to provide a qualitative comparison across sub-categories for each MLLM, as the corresponding tables (Tables 2 and 3) present complex numerical results that are not easy to interpret. We agree to move these figures to the appendix in the final version to make space for additional content in the main paper.
>
> **Response to Weakness 3-5:** Thank you for pointing out the typos and term misuses in our paper (Black Box => Closed-Source, LVLM => MLLM, and SOTA => SoTA). We will correct them in the final version.
>
> **Response to Question 1:**
> Our task involves using MLLMs for video understanding and question answering, specifically targeting physics and commonsense abnormalities. As a novel task, there are no existing datasets addressing this specific setting, video QA on physics violations in synthetic videos. Our paper is the first to introduce this problem, release a benchmark, and conduct both evaluation and fine-tuning experiments. Thus, it's not possible for us to provide evaluation on other datasets over exactly the same task.
>
> Importantly, our fine-tuning experiments aim to demonstrate the feasibility of the benchmark and point toward future directions for improving MLLMs. We are also happy to conduct additional experiments on  **relevant** benchmarks, such as MMVU [1] and MVBench [2], using our fine-tuned models to show that our training approach maintains generalization on broader video QA tasks. The evaluation results are presented below, using model names consistent with those in the original paper. **Refer to Table B in reviewer tQNG's console**
>
> Results on real-world datasets show that models fine-tuned with curriculum learning, whether using a composed or single training dataset for RFT, show little performance difference. Notably, the GRW-PRW-SR-R1-7B model, which is fully fine-tuned with all three datasets in a curriculum learning setup, achieves performance close to the model trained solely on real-world videos. Given its state-of-the-art results on synthetic video understanding and physics comprehension, this highlights the importance of incorporating real-world videos in RFT post-training to avoid performance degradation on general video tasks when using physics-focused data.
>
> To evaluate model feasibility beyond our current SoTA open-source model (Qwen2.5-VL-7B), we also present overall results on InternVL3-9B below, using our latest synthetic video dataset.
>
> | Metric   | InternVL3-9B-R1-CL | Qwen2.5-VL-7B-R1-CL | InternVL3-9B | Qwen2.5-VL-7B |
> |--|-|-|--|--|
> | Overall  | 58.2  | **58.7**     | 46.4         | 51.0           |
>
> According to the table, both RFT and curriculum learning yield notable improvements across all VideoHallu subcategories when performing RFT on both base MLLMs, demonstrating the generalizability of our training methods across MLLMs.
>
> [1] Zhao, Yilun, et al. "Mmvu: Measuring expert-level multi-discipline video understanding." Proceedings of the Computer Vision and Pattern Recognition Conference. 2025.
>
> [2] Li, Kunchang, et al. "Mvbench: A comprehensive multi-modal video understanding benchmark." Proceedings of the IEEE/CVF Conference on Computer Vision and Pattern Recognition. 2024.
>
> **Response to Question 2:**
>
> We present comprehensive results on our curriculum learning setup and training data compositions. Due to rebuttal word limits, detailed evaluations are provided in our response in **reviewer tQNG's console**. Our findings show that curriculum learning significantly outperforms baselines using only one or two datasets for RFT. While RFT on a single dataset can improve performance on specific tasks, it often harms performance on others. Moreover, for complex tasks requiring high-level comprehension, direct RFT without intermediate steps may be ineffective. Curriculum learning enables MLLMs to tackle such tasks more efficiently through gradual, step-by-step learning.
>
> We also run mixed data training without curriculum learning and show that it has worse results (52.3%) than training just on synthetic data (53.4%) or training them with curriculum learning (58.7%).
>
> Meanwhile, we are actively running more experiments comparing the MLLM's performance after RFT by composed data across all three datasets and each identical dataset under Training Separately setting. Please stay tuned for more results over the discussion period.
>
>
> **Response to Question 3:**
> Thank you for highlighting this weakness. Below are the implementation details:
> * Avg. frames per video: 96.0
> * Avg. video length: ~5.3 seconds
> * Avg. framerate (FPS): 23
> * Avg. frame resolution: 1042 × 588
> * Number of video generation models: 7
> * Avg. videos per model: 132
> * Maximum number of frames to sample for training: 18
> * Number of training iterations: 3 epochs
> * Maximun number of frames to sample for testing: 128
>
> **Response to Question 4:** The problem formulation section models the hallucination triggering mechanism in MLLMs, not just data collection. It outlines how both human annotators and MLLMs can generate QA pairs for synthetic videos with physics or commonsense violations, following the setting in Figure 1.
>
> The mapping function $T(\cdot)$ represents a question-generation mechanism that, given an entity $C$ (e.g., a watermelon in Figure 1), produces either a natural language question $T(Q)$ (e.g., In the video, the watermelon breaks in the middle. Is it intact or broken at the end?) or a textual description $T(C)$ of the event involving that entity (e.g., A bullet shot into the watermelon), possibly involving physics or commonsense violations.
>
> In practice, $T(\cdot)$ can be executed by human annotators, who identify a context $C$ in video $V$, then formulate both $T(Q)$ and $T(C)$. Alternatively, MLLMs can generate these when provided with a known violation. **While MLLMs cannot yet fully automate this process, they show promise in crafting QA pairs for synthetic videos when supplied with abnormal contexts, effectively serving as $T(\cdot)$.**
>
> The first constraint, $d[f_\mathtt{LLM}(\mathit{T}({\mathcal C}), \mathit{T}({\mathcal Q})), f_\mathtt{Human}(\mathit{T}({\mathcal C}), \mathit{T}({\mathcal Q}))]\leq \epsilon$, ensures that when only textual context $T(C)$ and question $T(Q)$ are given, the LLM’s response aligns semantically with human commonsense or physics knowledge. The second constraint, $d[f_\mathtt{Human}(\mathit{T}({\mathcal C}), \mathit{T}({\mathcal Q})), f_\mathtt{Human}(V, \mathit{T}({\mathcal Q}))]\geq \delta$, requires that hallucination cases lead to a significant difference in responses when humans view the actual video $V$, indicating a semantic gap of at least $\delta$. We measure these similarities using GPT-4 as an LLM-based evaluator. When framed as a binary classifier (same vs. different), both thresholds are implicit but can be estimated with high confidence given sufficient data.
>
>
> Examples are provided in Appendix C. Overall, our goal is to rigorously model hallucination in synthetic video understanding and guide future MLLM improvements.
>
>
> **Response to Question 5:**
> Our contribution goes beyond merely proposing a benchmark. We uncover a novel problem in synthetic video understanding: hallucinations in detecting physics and commonsense violations. In addition to introducing the benchmark and evaluating state-of-the-art MLLMs, we propose a curriculum learning-based training method to mitigate this issue. Our results show that even a small amount of synthetic video QA data can effectively help models improve in synthetic video understanding, suggesting a promising direction for future synthetic video hallucination mitigation.
>
> Moreover, our curriculum pipeline offers an alternative approach to fine-tuning MLLMs on complex tasks using limited, high-cost data by leveraging more accessible, simpler datasets to guide training. These contributions align with the Deep Learning sub-track of NeurIPS (e.g., architectures, generative models, optimization, foundation models, LLMs), which we selected upon submission.
>
> Regarding NeurIPS data/code disclosure policy, we were finalizing cleanup to prevent identification or double-blind policy violations at submission time. We will published our dataset and code for full reproducibility and include the link in the camera-ready version, **as we are not allowed to including any link in our NeurIPS rebuttal.** All necessary implementation details are provided.
>
> **Response to Question 6:** We are using 8 A100 (80GB) for our SFT, GRPO fine-tuning and evaluation inference. We are using Video-R1 codebase [1]. We will include the computation resource in the final version. Thank you for pointing out this issue.
>
> [1] Feng, Kaituo, et al. "Video-r1: Reinforcing video reasoning in mllms." arXiv preprint arXiv:2503.21776 (2025).
>
>
> **Response to Limitation 1:**  Our main limitation lies in the benchmark’s scalability, as both generating high-quality annotations and fine-tuning MLLMs at scale are expensive. The benchmark data were collected via paid APIs and expert-level human annotations, which are costly and time-consuming. Once more data gather and annotated with either human or MLLM automatic pipeline, future MLLMs are less likely hallucinated when probing physics/commonsense violations in synthetic videos. Another limitation, as noted by Reviewer DNqt, is the limited coverage of MLLMs used for fine-tuning. While we included InternVL3-8B fine-tuning results in our rebuttal, we are happy to expand the experiments over both currrent open-source video understaning MLLMs in the final version.
>
> We will also elaborate further on these limitations in the Appendix.

---

> > ### Author Response · Authors · 2025-08-01
> > **Added Experiment Table Results to demonstrate curriculum learning effectiveness**
> >
> > Table A. Ablation Study on Curriculum RFT.
> > | Metric      | GRW-PRW-SR-R1-7B  | GRW+SR-R1-7B | PRW+SR-R1-7B | Qwen2.5-VL-7B | Qwen2.5-VL-7B-RFT-Mixture-Data-Shuffle | Qwen2.5-VL-7B SFT |
> > |-------------|-----------------------------|--------------|--------------|---------------|----------------------------------------|-------------------|
> > | A-EC        | **58.8**                    | 51.3         | 50.7         | 53.5          | 52.4                                   | 57.7              |
> > | A-EP        | 62.4                        | 59.8         | 58.7         | **63.1**      | 58.9                                   | 62.2              |
> > | A-ERAC      | **69.8**                    | 68.9         | 67.7         | 65.5          | 66.8                                   | 61.0              |
> > | A-SR        | 59.6                        | 51.2         | 53.5         | 54.8          | 61.5                                   | **62.5**          |
> > | SC-CD       | 57.5                        | 53.5         | 55.2         | 47.6          | **61.1**                               | 51.3              |
> > | SC-SD       | **58.5**                    | 42.1         | 47.4         | 42.3          | 50.7                                   | 50.7              |
> > | SC-TD       | 61.1                        | 47.6         | **62.0**     | 43.9          | 56.1                                   | 53.1              |
> > | CS-K        | 54.2                        | 57.7         | **61.6**     | 50.8          | 57.0                                   | 55.1              |
> > | CS-R        | 58.8                        | 56.4         | **62.1**     | 42.0          | 53.9                                   | 52.9              |
> > | P-C         | **61.9**                    | 56.8         | 56.8         | 42.9          | 47.6                                   | 45.2              |
> > | P-CAP       | **60.0**                    | **60.0**     | 56.9         | 58.8          | 55.7                                   | 55.7              |
> > | P-M         | **59.3**                    | 46.2         | 44.8         | 45.6          | 57.7                                   | 48.5              |
> > | P-ST        | 61.6                        | 60.0         | **80.0**     | 40.0          | 55.0                                   | 45.0              |
> > | Overall     | **58.7**                    | 52.1     | 54.2     | 51.0          | 52.3                                   | 50.9              |
> >
> >
> > Table B. Real World Data Evaluation
> > | Model             | MMVU  | MVBench |
> > |------------------|-------|---------|
> > | Qwen-2.5VL-7B     | 58.7 | 69.6    |
> > | SFT (synthetic)   | 57.9  | 69.6    |
> > | SFT (GRW)         | 60.1  | 70.1    |
> > | SFT (PRW)         | 57.9  | 69.9   |
> > | GRW-R1-7B         | **61.3**  | 70.7    |
> > | PRW-R1-7B         | 58.3  | 69.7    |
> > | SR-R1-7B          | 59.1  | 69.3    |
> > | GRW-SR-R1-7B      | **61.3**  | **70.9**    |
> > | PRW-SR-R1-7B      | 59.1  | 69.7    |
> > | GRW-PRW-SR-R1-7B (Qwen2.5-VL-7B-R1-CL)      | **61.3**  | 70.0    |

---

> > ### Comment · Reviewer_DNqt · 2025-08-01
> >
> > Thank you for your response.
> >
> > I have reviewed the other reviewers’ comments as well as the authors’ replies. While some of my questions have been addressed, I believe others still warrant further clarification.
> >
> > As a general note, **I kindly ask the authors to be more specific in their responses.** For any revisions that do not involve plots or figures, it is insufficient to write statements such as *“we will include the computation resource in the final version”* or *“we will elaborate further on this.”* **Please clearly indicate what specific details will be added or changed**, as these revisions will form part of the final version of the paper if accepted.
> >
> > For instance, when you write *“We agree to move these figures to the appendix in the final version to make space for additional content in the main paper"*, please **specify exactly what content you intend to move and what you plan to add** in its place. Without concrete information, responses come across as vague and give the impression that they are meant to appease the reviewers rather than reflect a genuine commitment to revision.
> >
> > **AC may correct me if I’m mistaken, but I believe the following statement:**
> >
> > > Meanwhile, we are actively running more experiments comparing the MLLM's performance after RFT by composed data across all three datasets and each identical dataset under Training Separately setting. Please stay tuned for more results over the discussion period.
> >
> > **is not in line with the guidelines.** The rebuttal phase is intended to provide direct and complete responses to the reviewers' comments based on the material available at that time. It is not meant as an opportunity to defer answers to a later stage.
> > As far as I understand, I am not permitted to revise my evaluation based on new experiments that are shared after the rebuttal deadline, as this would not be fair to other submissions that adhered to the timeline.
> >
> > About **Q1**, if I understand correctly, the table that you are reporting corresponds to InternVL3-9B with and without curriculum learning. However, **the setup remains unclear to me.** To properly assess whether your current statements are supported, I would like to see the finalized version of Table 3 as you intend to include it in the paper.
> >
> > Specifically, it is unclear to me what R1-CL corresponds to (GRW+SR or PRW+SR or a different setting?). Additionally, I don’t understand why these results are based on InternVL3-9B, given that you already reported a zero-shot baseline using InternVL3-14B. Please clarify the motivation behind switching models and explain how this comparison is structured.
> >
> >
> > Also, I believe the 'InternVL3-8B' is a typo in the response. I guess you evaluate the InternVL3-9B model.

---

> ### Author Response · Authors · 2025-08-02
> **Re: Official Comment by Reviewer DNqt (1/3)**
>
> **Response to Follow-up 1:** Thank you for pointing out this issue. We would like to clarify that, in accordance with NeurIPS 2025 guidelines, authors are not permitted to submit external links, upload additional files that may risk identity disclosure, or modify the originally submitted PDF during the rebuttal phase. Therefore, we cannot specify what exact content will be added, revised, or removed in the camera-ready version, especially for non-technical aspects such as paper organization, figure replacement, or minor terminology inconsistencies, while the review and discussion process is still ongoing. These types of adjustments are intended to be addressed during the camera-ready submission phase.
>
> Regarding your follow-up questions, we plan to move Figures 3 and 5 to the appendix. In addition, we will include results from ablation studies and comparisons across various settings, such as SFT vs. GRPO, different GRPO and curriculum learning configurations, and data organization strategies (random shuffle vs. curriculum learning), all of which are included in Table A and C. We will also present generalization results on both Qwen2.5-VL-7B and InternVL3-9B, with and without curriculum learning. Furthermore, we will add evaluations on real-world relevant benchmarks, including MMVU and MVBench, which will be summarized in the updated Table B.
>
> **Response to Follow-up 2:**  In accordance with the NeurIPS rebuttal guidelines for authors
> > 1. Past experience suggests that effective responses focus on factual errors in the reviews and on responding to specific questions posed by the Reviewers. Your response is optional and should be reserved for cases when a response is called for. The response you submit by July 30th should contain all your arguments regarding the reviews. The discussion with reviewers should be reserved only for discussing these points rather than for raising new arguments.
>
> Our mention of running new experiments to **support our argument to Question 2** is  **not** intended to introduce new arguments in response to the Official Review of Submission6899 by Reviewer DNqt. Rather, it is meant to **reinforce and clarify our existing arguments** for the purpose of ongoing discussion.
>
> **Response to Follow-up 3:**  Here is the updated version of Table 3. Due to Markdown formatting constraints, we have separated the original composite table into four distinct sub-tables, now labeled Table C-1 through Table C-4. All models presented in Tables C-2, C-3, and C-4 are fine-tuned using Qwen2.5-VL-7B as the base model.
>
> To better organize the discussion on model generalization across different backbones (i.e., Qwen2.5-VL-7B and InternVL3-9B), we will include those results separately in Table B, along with a dedicated subsection for generalization analysis in the camera-ready version of the paper.
>
> Please note that due to the 10,000-character limit in the initial rebuttal, we were unable to include Table C as part of our previous submission. Including it would have exceeded the allowed length, given the scope of the initial reviews and the content already included. We plan to incorporate these additional results in the appendix or main paper as appropriate in the camera-ready version.

---

> > ### Author Response · Authors · 2025-08-02
> > **Re: Official Comment by Reviewer DNqt (2/3)**
> >
> > **Table C. Fine-Tuned Model Evaluation on VideoHallu (Updated Version)**
> > **Table C-1. MLLMs: Previous SoTA**
> > | Model                 | A-EC  | A-EP  | A-ERAC | A-SR  | SC-CD | SC-SD | SC-TD | CS-K  | CS-R  | P-C   | P-CAP | P-M   | P-ST  | Overall |
> > |----------------------|-------|-------|--------|-------|-------|-------|-------|-------|-------|-------|--------|-------|--------|---------|
> > | GPT-4o               | 45.4  | 58.1  | 61.5   | 45.2  | 40.5  | 35.9  | 37.3  | 54.1  | 34.8  | 47.6  | 47.1  | 46.7  | 50.0  | 45.5    |
> > | InternVL3-9B                   | 45.9  | 55.7  | 58.0   | 40.5  | 40.5  | 37.3  | 42.9  | 50.8  | 39.1  | 47.6  | 29.4  | 46.7  | 50.0  | 46.4    |
> > | InternVL3-14B        | 49.2  | 53.2  | 57.5   | 50.0  | 42.9  | 37.3  | 42.9  | 44.3  | 40.6  | 38.1  | 47.1  | 47.8  | 60.0  | 46.7    |
> > | Gemini-2.5-Pro       | 56.8 | 61.6  | 65.5   | 57.1 | 50.0  | 41.5  | 40.9  | 52.5  | 46.4  | 38.1  | 47.1  | 35.6  | 40.0  | 49.8    |
> > | Video-R1             | 51.4  | 62.1  | 67.8   | 35.7  | 40.5  | 45.1  | 43.7  | 52.5  | 46.4  | 38.1  | 58.8  | 50.0  | 60.0  | 50.8    |
> > | Qwen2.5-VL-7B        | 53.5  | **63.1** | 65.5   | 54.8  | 47.6  | 42.3  | 43.9  | 50.8  | 42.0  | 42.9  | 58.8  | 45.6  | 40.0  | 51.0    |
> > | Qwen2.5-VL-32B       | 54.6  | 60.1  | 67.8   | 52.4  | **59.5** | 42.3  | 40.7  | 55.7  | 58.0  | 42.9  | 52.9  | 51.1 | 40.0  | 51.4 |
> >
> > **Table C-2. Additional Strategies**
> > | Model                 | A-EC  | A-EP  | A-ERAC | A-SR  | SC-CD | SC-SD | SC-TD | CS-K  | CS-R  | P-C   | P-CAP | P-M   | P-ST  | Overall |
> > |----------------------|-------|-------|--------|-------|-------|-------|-------|-------|-------|-------|--------|-------|--------|---------|
> > | Mixture-Data-Shuffle | 52.4  | 58.9  | 66.8   | 61.5  | 61.1  | 50.7  | 56.1  | 57.0  | 53.9  | 47.6  | 55.7  | 57.7  | 55.0  | 52.3    |
> > | SFT | 57.7  | 62.2  | 61.0   | 62.5  | 51.3  | 50.7  | 53.1  | 55.1  | 52.9  | 45.2  | 55.7  | 48.5  | 45.0  | 50.9    |
> >
> > **Table C-3. Training Separately**
> > | Model                 | A-EC  | A-EP  | A-ERAC | A-SR  | SC-CD | SC-SD | SC-TD | CS-K  | CS-R  | P-C   | P-CAP | P-M   | P-ST  | Overall |
> > |----------------------|-------|-------|--------|-------|-------|-------|-------|-------|-------|-------|--------|-------|--------|---------|
> > | GRW-R1-7B            | 48.7  | 60.3  | 67.5   | 34.9  | 51.7  | 49.1  | 42.9  | 57.7  | 60.0  | 62.2  | 56.9  | 42.7  | 50.0  | 51.5    |
> > | PRW-R1-7B            | 49.3  | 60.3  | 67.1   | 41.9  | 55.2  | 43.9  | 57.1  | 57.7  | 60.0  | **64.9** | 54.9  | 41.3  | 70.0  | 52.2    |
> > | SR-R1-7B             | 50.7  | 58.7  | 68.3   | 48.8  | 56.9  | 50.9 | **66.7** | 61.5  | 61.4  | 54.1  | 56.9  | 43.4  | 70.0  | 53.4 |
> >
> > **Table C-4. Curriculum Learning**
> > | Model                 | A-EC  | A-EP  | A-ERAC | A-SR  | SC-CD | SC-SD | SC-TD | CS-K  | CS-R  | P-C   | P-CAP | P-M   | P-ST  | Overall |
> > |----------------------|-------|-------|--------|-------|-------|-------|-------|-------|-------|-------|--------|-------|--------|---------|
> > | GRW+SR-R1-7B         | 51.3  | 59.8  | 68.9 | 51.2  | 53.5  | 42.1  | 47.6  | 57.7  | 56.4  | 56.8  | **60.0** | 46.2  | 60.0  | 52.1    |
> > | PRW+SR-R1-7B         | 50.7  | 58.7  | 67.7   | 53.5  | 55.2  | 47.4  | 62.0  | **61.6** | **62.1** | 56.8  | 56.9  | 44.8  | **80.0** | 54.2 |
> > | **Qwen2.5-VL-7B-R1-CL**    | **58.8** | 62.4 | **69.8** | **59.6**  | 57.5  | **58.5**  | 61.1  | 54.2  | 58.8  | 61.9  | **60.0**  | **59.3**  | 61.6  | **58.7** |

---

> > > ### Author Response · Authors · 2025-08-02
> > > **Re: Official Comment by Reviewer DNqt (3/3)**
> > >
> > > **Response to Follow-up 4:** Thank you for pointing this out. To clarify, R1-CL is a shorthand for GRW-PRW-SR-R1-7B, which follows a curriculum learning schedule: training sequentially on General Real-World (GRW), then Physics Real-World (PRW), and finally Synthetic Reasoning (SR) data. Given the length of the full name, we use the abbreviation R1-CL as a suffix for clarity and readability. For example, we refer to models as Qwen2.5-VL-7B-R1-CL or InternVL3-9B-R1-CL, rather than the more verbose Qwen2.5-VL-GRW-PRW-SR-R1-7B.
> > >
> > > In Table 2 of our paper, we report results for **InternVL3-9B (row 4 in MLLM sub-section, 7th row from the top of Table 2)**, which we include as a baseline for comparison. To ensure fair generalization comparisons across models of similar size, we primarily use InternVL3-9B. The only reason we initially presented InternVL3-14B in Table 3 was because it achieved the best performance within the InternVL3 family, according to our evalution results in Table 2. In response to your feedback, we have updated Table 3 accordingly and provided the revised version here for further discussion.
> > >
> > > **Response to Follow-up 5:**  Thank you for pointing this out. As previously mentioned, all our experiments related to model generalization (as reflected in the updated version of Table 3) are conducted using InternVL3-9B. You are correct, this was a typo on our part.

---

> > > > ### Comment · Reviewer_DNqt · 2025-08-02
> > > >
> > > > I am well aware of the guidelines, but thanks for reminding me. To clarify, I wasn't requesting a camera-ready version; I simply wanted a clear and detailed description of the potential changes, similar to what you're addressing in this phase of the discussion. To conclude, thank you for your clarification. I will discuss my final rating with the other reviewers.

---

### Official Review · Reviewer_bcUY · 2025-07-03

**Clarity:** 3
**Significance:** 3
**Originality:** 3
**Rating:** 4
**Confidence:** 3

**Summary:**

The paper presents VideoHallu which is a benchmark to evaluate the critical (physics-based) thinking abilities of the multi-modal large language models (MLLMs). By using 3000 synthetic videos paired with counterintuitive QA, it evaluates and mitigate hallucinations in multi-modal large language models (MLLMs) where the generations can show abnormal or abnormal or counterintuitive visual phenomena (such as an object exploding and reassembling back). The paper presents an exhaustive evaluation strategy into multiple categories such as spatial-temporal inconsistency, physics violations, and commonsense inconsistencies. Multiple state-of-the-art MLLMs are evaluated based on these metrics and the paper discusses several insights such as the MLLMs fail to answer counterintuitive questions, chain of thought also underperforms, and solely based on real-world data can create a bias of how the real-world works and limiting the understanding of counterintuitive physical phenomenon.

To address this, the paper proposes a curriculum learning-based finetuning strategy using general videos, real-world physics videos, and synthetic video QA data to improve physical understanding of the models. The evaluation shows modest gains in performance through this approach.

**Questions:**

Will the VideoHallu dataset be publicly released with all associated QA pairs and metadata?

**Ethical Concerns:**

["NO or VERY MINOR ethics concerns only"]

**Final Justification:**

During the rebuttal phase, the authors addressed my concern of whether the curriculum training is helpful on all evaluation axes (alignment, spatio-temporal consistency, and physics) by showing exhaustive results. I still have some concern about the data coverage and diversity which I understand wouldn't have been possible to address during the rebuttal phase. Therefore, I would prefer to maintain my rating to borderline accept. The authors are encouraged to explore diversity in their dataset.

**Limitations:**

Not sure. The question 10: broader impact of the checklist has answer as [NA].

**Quality:**

3

**Strengths And Weaknesses:**

Strengths:
1. The idea of using counterintuitive examples (such as bullet being shot into a watermelon) is interesting.
2. The paper is well-written and clearly explains how synthetic data can be helpful to provide counterintuitive examples and thus ensure physical plausibility.
3. The paper presents an exhaustive set of categories and subcategories for evaluation of the physics on the basis of alignment, spatial-temporal consistency, common-sense reasoning, and physics.

Weakness:
While the idea of the paper of using counterintuitive questions/examples via synthetic data is sound, the results from the proposed contributions of curriculum training with general videos+real-world videos+synthetic videos and using GRPO in Table 3 doesn't seem to add much strength to the contributions and don't show much performance improvement on the proposed evaluation categories of alignment, spatial-temporal consistency, and physics. While the overall score is in favor of curriculum training using physics-real world and synthetic data, if we look at the results for each category in Table 3, for example in physics, for some metrics, training separately works better and for some metrics, curriculum training works better. The inconsistency raises some doubt as to whether curriculum learning is helpful or can training separately can be helpful in general. Similarly in alignment, the previous SoTA methods seem to show better performance majorly. Can the authors discuss why the performance is still on lower end when compared to previous SoTA methods?

---

> ### Author Rebuttal · Authors · 2025-07-31
>
> Thank you for the thoughtful feedback. We're glad you enjoyed the paper and found the use of counterintuitive examples compelling. We truly appreciate your recognition of the writing clarity and the role of synthetic data in promoting physical plausibility.
>
> **Response to Weakness 1:**
> Thank you for your insightful doubts. Aas the results shown in Table A, We train Qwen2.5-VL-7B on mixed data without performing curriculum learning on physics + synthetic video data (Qwen2.5-VL-7B-RFT-Mixture-Data-Shuffle) and show that no curiculum learning has an overall accuracy of 52.3%, where training on physics only data has 52.2%, and 53.4% training on synthetic data. Combining them without curriculum learning leads to minimal performance improvement or even degradation than learning first from easy real world physics data then transition to the harder synthetic video data (54.2%). Learning merely from mixed data without increasing difficulty level would cause the model to generate wrong responses for the more challenging data thus not backpropagating on the synthetic videos.
>
> **Response to Weakness 2:**  Thank you for pointing out this limitation. We agree that in the current version, our reinforcement fine-tuning (RFT) with limited data improves performance on physics and commonsense reasoning but at the cost of reduced performance on alignment-related QA in VideoHallu. We attribute this trade-off to the narrow focus of our RFT dataset, which emphasizes video understanding (real and synthetic) while lacking coverage on alignment tasks. In future work, we plan to incorporate more diverse and balanced training data to enhance MLLMs’ physics/commonsense reasoning without compromising performance on alignment QA. This includes expanding data coverage and improving training efficiency.
>
> **Response to Question 1:**
> We will release our dataset and code in the final version as one of the core contribution of our paper. Also, we are happy to provide the metadata of our dataset as below. Here are more details of the metadata that will be included in the script:
> * Avg. frames per video: 96.0
> * Avg. video length: ~5.3 seconds
> * Avg. framerate (FPS): 23
> * Avg. frame resolution: 1042 × 588
> * Number of video generation models: 7
> * Avg. videos per model: 132
> * Maximum number of frames to sample for training: 18
> * Number of training iterations: 3 epochs
> * Maximun number of frames to sample for testing: 128
>
> All video question-answer (QA) pairs used in our benchmark will be released alongside the final version of our paper.
>
>
> **Response to Limitaion 1:** According to NeurIPS guidelines, Question 10 (potential negative societal impacts) is out of scope for our work. We propose a benchmark focused on physics and commonsense violations in synthetic video QA tasks and demonstrate its utility in improving foundation models’ video understanding through reinforcement learning fine-tuning. Our work does not involve negative societal impacts, as it aims to mitigate such issues in other systems.

---

> > ### Author Response · Authors · 2025-08-01
> > **Added Experiment Table Results to demonstrate curriculum learning effectiveness**
> >
> > Table A. Ablation Study on Curriculum RFT.
> > | Metric      | GRW-PRW-SR-R1-7B  | GRW+SR-R1-7B | PRW+SR-R1-7B | Qwen2.5-VL-7B | Qwen2.5-VL-7B-RFT-Mixture-Data-Shuffle | Qwen2.5-VL-7B SFT |
> > |-------------|-----------------------------|--------------|--------------|---------------|----------------------------------------|-------------------|
> > | A-EC        | **58.8**                    | 51.3         | 50.7         | 53.5          | 52.4                                   | 57.7              |
> > | A-EP        | 62.4                        | 59.8         | 58.7         | **63.1**      | 58.9                                   | 62.2              |
> > | A-ERAC      | **69.8**                    | 68.9         | 67.7         | 65.5          | 66.8                                   | 61.0              |
> > | A-SR        | 59.6                        | 51.2         | 53.5         | 54.8          | 61.5                                   | **62.5**          |
> > | SC-CD       | 57.5                        | 53.5         | 55.2         | 47.6          | **61.1**                               | 51.3              |
> > | SC-SD       | **58.5**                    | 42.1         | 47.4         | 42.3          | 50.7                                   | 50.7              |
> > | SC-TD       | 61.1                        | 47.6         | **62.0**     | 43.9          | 56.1                                   | 53.1              |
> > | CS-K        | 54.2                        | 57.7         | **61.6**     | 50.8          | 57.0                                   | 55.1              |
> > | CS-R        | 58.8                        | 56.4         | **62.1**     | 42.0          | 53.9                                   | 52.9              |
> > | P-C         | **61.9**                    | 56.8         | 56.8         | 42.9          | 47.6                                   | 45.2              |
> > | P-CAP       | **60.0**                    | **60.0**     | 56.9         | 58.8          | 55.7                                   | 55.7              |
> > | P-M         | **59.3**                    | 46.2         | 44.8         | 45.6          | 57.7                                   | 48.5              |
> > | P-ST        | 61.6                        | 60.0         | **80.0**     | 40.0          | 55.0                                   | 45.0              |
> > | Overall     | **58.7**                    | 52.1     | 54.2     | 51.0          | 52.3                                   | 50.9              |
> >
> >
> > Table B. Real World Data Evaluation
> > | Model             | MMVU  | MVBench |
> > |------------------|-------|---------|
> > | Qwen-2.5VL-7B     | 58.7 | 69.6    |
> > | SFT (synthetic)   | 57.9  | 69.6    |
> > | SFT (GRW)         | 60.1  | 70.1    |
> > | SFT (PRW)         | 57.9  | 69.9   |
> > | GRW-R1-7B         | **61.3**  | 70.7    |
> > | PRW-R1-7B         | 58.3  | 69.7    |
> > | SR-R1-7B          | 59.1  | 69.3    |
> > | GRW-SR-R1-7B      | **61.3**  | **70.9**    |
> > | PRW-SR-R1-7B      | 59.1  | 69.7    |
> > | GRW-PRW-SR-R1-7B (Qwen2.5-VL-7B-R1-CL)      | **61.3**  | 70.0    |

---

> ### Author Response · Authors · 2025-08-05
> **Friendly Reminder Regarding NeurIPS Rebuttal – 48 Hours Left**
>
> Dear Reviewer bcUY,
>
> We appreciate the time and effort you've dedicated to reviewing our work. Your insights have been extremely useful for us as they will make our work better, and we have taken care to address each concern raised. If there are any additional points of discussion, we would be grateful for the opportunity to engage further.
>
> Since there are only 48 hours remaining in the rebuttal period, we kindly remind you to reconsider the evaluation of our work in light of the revisions and clarifications provided.
>
> Regards,
>
> Authors

---

> > ### Comment · Reviewer_bcUY · 2025-08-07
> > **Rebuttal Acknowledgement**
> >
> > Thank you authors for the detailed rebuttal and for presenting the detailed experimental evaluation. My primary concerns were that the proposed contribution of curriculum training doesn't show much performance improvement and that there is some inconsistency in the performance gains when model is trained separately and with curriculum training. The authors pointed this to the limitation in their dataset which is diverse enough for video understanding but lacks coverage for alignment. The authors plan to expand their data coverage and diversity. Reading through the rebuttal, while the idea of counterintuitive questions/examples to evaluate the physics-based thinking abilities of MLLMs is sound, it seems like the dataset still requires some work to verify if the curriculum training is helpful on all evaluation axes (alignment, spatio-temporal consistency, and physics). Taking into account the rebuttal, I will discuss with the other reviewers' to update my final score.

---

> > > ### Author Response · Authors · 2025-08-08
> > > **Re: Rebuttal Acknowledgement by Reviewer bcUY**
> > >
> > > Thank you for your constructive feedback. To further substantiate our claims, we present supplementary category-wise results for the models in Table A, showing how our training strategy helps to improve performance across all evaluated dimensions.
> > >
> > > Table D: Category-wise Results for Table A
> > > | Model | Alignment | Spatial–temporal Consistency | Common Sense | Physics | Overall |
> > > |-------|-----------|------------------------------|--------------|---------|---------|
> > > | GRW-PRW-SR-R1-7B | 62.8 | 59.0 | 56.5 | 60.7 | 58.7 |
> > > | GRW+SR-R1-7B | 57.8 | 47.7 | 57.1 | 55.9 | 52.1 |
> > > | PRW+SR-R1-7B | 57.7 | 54.9 | 61.9 | 59.6 | 54.2 |
> > > | Qwen2.5-VL-7B-RFT-Mixture-Data-Shuffle | 59.9 | 56.0 | 55.5 | 54.0 | 52.3 |
> > > | Qwen2.5-VL-7B SFT | 60.7 | 51.7 | 54.0 | 48.6 | 50.9 |
> > > | Qwen2.5-VL-7B | 59.2 | 44.6 | 46.4 | 46.8 | 51.0 |
> > >
> > > Compared to the base model Qwen2.5-VL-7B, our final fine-tuned model GRW-PRW-SR-R1-7B achieves consistent gains across all categories:
> > >
> > > * Alignment: +3.4
> > > * Spatial–temporal Consistency: +14.4
> > > * Common Sense: +10.1
> > > * Physics: +13.9
> > >
> > > These results show that the proposed curriculum learning strategy does not merely overfit to a single evaluation dimension or dataset but generalizes improvements across all four dimensions. The largest gains occur in spatial–temporal consistency and physics, two of the most challenging aspects for current MLLMs. Even in the alignment category, where the base model, as the state-of-the-art, is relatively strong, our method still provides a notable +3.4% improvement.
> > >
> > > These findings reinforce that our training data and training strategy can lead to generalizable gains that extend beyond a single aspect, directly addressing the robustness and reasoning limitations of current models on synthetic video understanding and lack of synthetic video understanding in pretraining data.

---

### Official Review · Reviewer_tQNG · 2025-07-03

**Clarity:** 3
**Significance:** 3
**Originality:** 3
**Rating:** 5
**Confidence:** 3

**Summary:**

- This work introduces VideoHallu, a video hallucination benchmark containing 3K QA pairs for synthetic videos created by modern video generation models. The goal is

- They define four evaluation categories, then employ human experts to create 141 adversarial prompts with which they generate 987 videos across seven video generation models.

- Question answer pairs are then created for each video by human experts and categorized into sub-categories

- They use LLM as a judge for evaluation, finding that frontier MLLMs struggle with counterintuitive questions and counterfactual entity changes

- Reasoning models do not fare better on these synthetic videos, having been trained only on real-world videos, so treating real-world rules as universal

- Propose to improve reasoning on synthetic videos via a three-stage curriculum which trains on two real-world datasets followed by the synthetic VideoHallu. They use GRPO with Answer-Equivalence BERT to assess accuracy and a BERTScore-like cosine similarity for the reward signal when training Qwen 2.5 VL

- The proposed curriculum training, especially the physics subset, improves performance on their benchmark

**Questions:**

Please see the weaknesses. If these evaluation limitations are addressed I would be happy to raise my score

**Ethical Concerns:**

["NO or VERY MINOR ethics concerns only"]

**Final Justification:**

The authors have addressed my key concerns by training SFT models for comparison with RFT, training on a full data mixture to compare to their proposed curriculum, and by evaluating on real-world (non-synthetic) datasets. With these additional experiments the paper is more well-rounded. Therefore, I change my score to accept.

**Limitations:**

Yes

**Paper Formatting Concerns:**

No major concerns.

**Quality:**

3

**Strengths And Weaknesses:**

# Strengths

- Strong dataset contribution with human-crafted prompts and adversarial QA pairs organized into a taxonomy

- Important contribution in noting MLLMs’ deficiencies on synthetic videos, despite strong performance on real-world videos

- Evaluations show that the proposed training regime improves performance over the baseline model


# Weaknesses

- The benefits of the proposed curriculum and use of reinforcement finetuning are unclear. Therefore, one is unsure whether the gains should be attributed to the training data mixture or the training strategy

    - Curriculum learning is compared to training on single subsets. However, a better comparison would be to train the base model on a mixture of the three datasets to evaluate the usefulness of curriculum learning in this setting

    - There appears to be no SFT baseline, so GRPO’s usefulness here is unclear.

- Without knowing whether the training strategy makes a significant difference over batched SFT, the takeaways of Table 3 and Figure 5 seem to be reduced to “training on synthetic video data improves performance on a synthetic video benchmark over the base models”

- No evaluations on real-world video datasets to ensure regression does not occur after training on only synthetic video data

---

> ### Author Rebuttal · Authors · 2025-07-31
>
> Thank you for the thoughtful feedback. We appreciate the recognition of our human-crafted dataset with adversarial QA pairs and taxonomy, as well as our effort to expose MLLM weaknesses on synthetic videos. We are also glad our training approach was noted to improve performance over the baseline.
>
> Given the limitation on the rebuttal length, we include our all our supplementary experiment results in tables below, and we will cite these tables throught our rebuttal:
>
> **Table A. Ablation Study on Curriculum RFT**
> | Metric      | GRW-PRW-SR-R1-7B  | GRW+SR-R1-7B | PRW+SR-R1-7B | Qwen2.5-VL-7B | Qwen2.5-VL-7B-RFT-Mixture-Data-Shuffle | Qwen2.5-VL-7B SFT |
> |-------------|-----------------------------|--------------|--------------|---------------|----------------------------------------|-------------------|
> | A-EC        | **58.8**                    | 51.3         | 50.7         | 53.5          | 52.4                                   | 57.7              |
> | A-EP        | 62.4                        | 59.8         | 58.7         | **63.1**      | 58.9                                   | 62.2              |
> | A-ERAC      | **69.8**                    | 68.9         | 67.7         | 65.5          | 66.8                                   | 61.0              |
> | A-SR        | 59.6                        | 51.2         | 53.5         | 54.8          | 61.5                                   | **62.5**          |
> | SC-CD       | 57.5                        | 53.5         | 55.2         | 47.6          | **61.1**                               | 51.3              |
> | SC-SD       | **58.5**                    | 42.1         | 47.4         | 42.3          | 50.7                                   | 50.7              |
> | SC-TD       | 61.1                        | 47.6         | **62.0**     | 43.9          | 56.1                                   | 53.1              |
> | CS-K        | 54.2                        | 57.7         | **61.6**     | 50.8          | 57.0                                   | 55.1              |
> | CS-R        | 58.8                        | 56.4         | **62.1**     | 42.0          | 53.9                                   | 52.9              |
> | P-C         | **61.9**                    | 56.8         | 56.8         | 42.9          | 47.6                                   | 45.2              |
> | P-CAP       | **60.0**                    | **60.0**     | 56.9         | 58.8          | 55.7                                   | 55.7              |
> | P-M         | **59.3**                    | 46.2         | 44.8         | 45.6          | 57.7                                   | 48.5              |
> | P-ST        | 61.6                        | 60.0         | **80.0**     | 40.0          | 55.0                                   | 45.0              |
> | Overall     | **58.7**                    | 52.1     | 54.2     | 51.0          | 52.3                                   | 50.9              |
>
>
> **Table B. Real World Data Evaluation**
> | Model             | MMVU  | MVBench |
> |------------------|-------|---------|
> | Qwen-2.5VL-7B     | 58.7 | 69.6    |
> | SFT (synthetic)   | 57.9  | 69.6    |
> | SFT (GRW)         | 60.1  | 70.1    |
> | SFT (PRW)         | 57.9  | 69.9   |
> | GRW-R1-7B         | **61.3**  | 70.7    |
> | PRW-R1-7B         | 58.3  | 69.7    |
> | SR-R1-7B          | 59.1  | 69.3    |
> | GRW-SR-R1-7B      | **61.3**  | **70.9**    |
> | PRW-SR-R1-7B      | 59.1  | 69.7    |
> | GRW-PRW-SR-R1-7B (Qwen2.5-VL-7B-R1-CL)      | **61.3**  | 70.0    |
>
>
>
> **Response to Weakness 1:**
> The gain of the model improvement come from curricumum reinforcement fine-tuning strategy and augmenting the right training data. We provide more clear reasons and explaination below:
> 1. We present new results on non-curriculum learning by mixing physics and synthetic video data in Qwen2.5-VL-7B-RFT-Mixture-Data-Shuffle, yielding 52.3% accuracy, only 0.1% better than physics-only training, 1.1% lower than synthetic-only, and 1.9% below curriculum learning (PRW+SR-R1-7B). This suggests that just 0.8% of the gain comes from training strategy. In contrast, organizing the data using curriculum learning (GRW-PRW-SR-R1-7B) leads to a 7.7% improvement over the base model after RFT.
>
> 2. Our findings show that data mixing without curriculum learning achieves 52.3% accuracy, which is lower than the 54.2% from curriculum learning (PRW+SR-R1-7B).
>
> 3. From the GRW+SR-R1-7B results, we observe that mixing general world and synthetic data slightly degrades performance compared to synthetic-only training. However, integrating physics and synthetic data via curriculum learning yields superior results. Since the synthetic dataset only has 800 samples, augmenting it with physics data significantly boosts performance, highlighting the importance of strategic data combinations in training.
>
>
> **Response to Weakness 1.1**
> We present the results on training on training on the mixed dataset with shuffling in the result of Qwen2.5-VL-7B-RFT-Mixture-Data-Shuffle. Training on mixed data has an overall accuracy of 52.3%, where training on physics only data has 52.2%, and 53.4% training on synthetic data. Combining them without curriculum learning leads to minimal performance improvement or even degradation than learning first from easy real world physics data then transition to the harder synthetic video data (54.2%). Learning merely from mixed data without increasing difficulty level would cause the model to generate wrong responses for the more challenging data thus not backpropagating on the synthetic videos.
>
> **Response to Weakness 1.2:** There appears to be no SFT baseline, so GRPO’s usefulness here is unclear.
>
> We train the 7B model with SFT on our synthetic video understanding datasets and get the following results, with no significant improvement on our benchmark data. Since this is a challenging dataset, training the model with SFT does not lead to improvement (50.9% compared to base model 51%) because SFT simply maximizes likelihood of the ground-truth answers and does not expose the model to its own chain-of-thought. Synthetic video understanding tasks require temporal reasoning—tracking objects and causal events across frames—and SFT’s teacher-forcing objective doesn’t train the model to perform multi-step inference at test time. In addition, even the SoTA model has only 51% accuracy on the test data, if we use distilled CoT data to train the model, there will not be improvement since the model we are performing SFT/RFT is the SoTA model. Using RFT can fully leverage model's CoT process to generalize and improve temoral and physics reasoning on synthetic videos.
>
> **Response to weakness 2**: We present new results in our table and highlight the following key takeaways:
>
> **SFT vs. RFT:** Supervised fine-tuning (SFT) only optimizes the likelihood of ground-truth answers and does not involve the model's own chain-of-thought (CoT), limiting its ability to learn physics rules. In contrast, reinforcement fine-tuning (RFT) leverages the model’s CoT process, resulting in stronger performance gains on the synthetic benchmark.
>
> **Why Not Distill CoT from SoTA:** Distilling CoT data from a state-of-the-art (SoTA) model for SFT is ineffective in our case, as the SoTA model is Qwen2.5-VL-7B itself—the one being trained. This would mean training the model on answers it already knows, offering little benefit.
>
> RFT is more effective than SFT for improving synthetic video understanding in our setup, and we therefore adopt it as our primary training strategy.
>
> **Response to weakness 3**: We provided real-world video benchmark results (We used VLMEvalkit). We show that there is no big difference or degradation on real world datasets when training with synthetic data, where a very small fluctuation of accuracy (<0.5%) is expected after finetuning.

---

> ### Author Response · Authors · 2025-08-05
> **Friendly Reminder Regarding NeurIPS Rebuttal – 48 Hours Left**
>
> Dear Reviewer tQNG,
>
> We appreciate the time and effort you've dedicated to reviewing our work. Your insights have been extremely useful for us as they will make our work better, and we have taken care to address each concern raised. If there are any additional points of discussion, we would be grateful for the opportunity to engage further.
>
> Since there are only 48 hours remaining in the rebuttal period, we kindly remind you to reconsider the evaluation of our work in light of the revisions and clarifications provided.
>
> Regards,
>
> Authors

---

> > ### Comment · Reviewer_tQNG · 2025-08-05
> >
> > Thank you for your response and for the additional experiments to address my concerns. I assume the SFT model (Qwen2.5-VL-7B SFT) was trained for 1 epoch over the combined datasets (as the paper says the RFT models were trained for 1 epoch)?

---

> > > ### Author Response · Authors · 2025-08-05
> > > **About SFT Training Epoch Details**
> > >
> > > Thank you very much for your response. Yes. We train SFT for one epoch to keep it consistent as the RFT method for a fair comparison.

---

> > > > ### Comment · Reviewer_tQNG · 2025-08-05
> > > >
> > > > Understood, thank you. I will take these new experiments into account for my final score.

---

### Note · Authors · 2025-08-16

We sincerely thank all reviewers and Area Chairs for their time, effort, and insightful feedback throughout the review and rebuttal process.

As a final remark for Submission 6899, our responses and revisions primarily address the raised comments and identified weaknesses in the following aspects:

1. **Supporting Experiments:** including comparisons of training strategies (SFT vs. RFT, data shuffle vs. curriculum), curriculum strategy in the GRPO pipeline, base model choices, and benchmark comparisons with prior work.

2. **Dataset and Code Transparency:** providing dataset metadata, disclosure details, and computational resource considerations.

3. **Technical Clarifications:** further explanation of the curriculum GRPO strategy, our problem formulation, and contributions from a broader perspective.

4. **Minor Revisions:** improvements in grammar, typos, figure organization, and results presentation.

Throughout the discussion, we have carefully addressed all reviewer concerns under the regulation of NeurIPS 2025. We provided clarifications and additional experimental evidence, while we also outlined concrete plans for incorporating modifications in the camera-ready version.

We once again thank the reviewers and Area Chairs for their valuable feedback and engagement, which have greatly strengthened our work.

---

### Decision · Program_Chairs · 2025-09-17

**Decision:**

Accept (poster)

**Comment:**

This paper presents a video-QA benchmark to evaluate MLLM's capabilities to identify hallucinations in generated videos, as well as a training strategy to improve MLLM performance in such a setting.  The final reviews are mixed, with one accept, one borderline accept and one borderline reject. The main source of contention seems to be the experiments (or lack thereof) to strongly support the idea, especially with respect to the curriculum learning.

Having read the reviews, responses and reviewer discussion, the AC recommends the paper to be accepted. The benefits outweigh the weaknesses and the work will make for an interesting addition to the program. That having been said, the authors are kindly requested to incorporate their additional experiments and explanations throughout the discussion into the camera-ready.